# NNiT: Width-Agnostic Neural Network Generation with Structurally Aligned Weight Spaces

**Jiwoo Kim** [1]  **Swarajh Mehta** [2]  **Hao-Lun Hsu** [2]  **Hyunwoo Ryu** [3]  **Yudong Liu** [1]  **Miroslav Pajic** [1]

## Abstract

Generative modeling of neural network parameters is often tied to architectures because standard parameter representations rely on known weight-matrix dimensions. Generation is further complicated by permutation symmetries that allow networks to model similar input-output functions while having widely different, unaligned parameterizations. In this work, we introduce Neural Network Diffusion Transformers (NNiTs), which generate weights in a width-agnostic manner by tokenizing weight matrices into patches and modeling them as locally structured fields. We establish that Graph HyperNetworks (GHNs) with a convolutional neural network (CNN) decoder structurally align the weight space, creating the local correlation necessary for patch-based processing. Focusing on Multilayer Perceptrons (MLPs), where permutation symmetry is especially apparent, NNiTs generate fully functional networks across a range of architectures. Our approach jointly models discrete architecture tokens and continuous weight patches within a single sequence model. On ManiSkill3 robotics tasks, NNiT achieves $> 85\%$ success on architecture topologies unseen during training, while baseline approaches fail to generalize; the same pipeline also generalizes to MNIST classification beyond the robotic control setting.

## 1. Introduction

Diffusion Transformers have demonstrated scalability across various modalities, such as computer vision image generation (Peebles & Xie, 2022; Bao et al., 2023; Xie et al., 2025; 2024; Gao et al., 2023), video generation (Liu et al., 2025; Ho et al., 2022; Menapace et al., 2024; Chen et al., 2025; Ma et al., 2025; Gupta et al., 2023), 3D generation (Mo et al., 2023; Xiang et al., 2024; Wu et al., 2024), and protein or molecule design (Wu et al., 2022; Joshi et al., 2025; Liu et al., 2024; Li et al., 2025a). Recently, this generative paradigm has been applied to parameter synthesis, where models directly generate weights for functional neural networks (Liang et al., 2024; Soro et al., 2024; Erkoç et al., 2023). These approaches aim to sample full neural networks from a learned distribution, bypassing the computational cost of traditional training.

A central challenge in neural parameter synthesis is the permutation symmetry of network weights (Navon et al., 2023; Zhao et al., 2025), where many distinct parameterizations correspond to the same function, so adjacent weights are spatially uncorrelated. To address this challenge, recent approaches generate weights in a latent embedding (Soro et al., 2024; Saragih et al., 2025), or apply explicit canonicalization methods (Ainsworth et al., 2023) before flattening parameters into 1D vectors (Sch"urholt et al., 2024). While these approaches can successfully transfer across varying depths, they remain fragile to changes in width. Once a weight matrix is collapsed into a fixed-dimensional vector, the generative prior becomes coupled to the matrix size seen during training. Even with alignment, changing the layer width changes token dimensionality and disrupts learned correspondence, preventing generalization to geometries unseen during training.

Consequently, to resolve this limitation, we rethink the role of Graph HyperNetworks (GHNs) (Zhang et al., 2020; Knyazev et al., 2021) and apply them as both a data source generator and a mechanism for aligning weight space. GHNs propagate information over the architecture graph and generate layer parameters in fixed order, anchored at an input node with task features. In our implementation of the GHN, we use a CNN decoder, which imposes an explicit locality bias in the weight-space. This results in weights with consistent local spatial correlations across the population. On the other hand, the standard Stochastic Gradient Descent (SGD) typically produces functionally equivalent

[1]Department of Electrical and Computer Engineering, Duke University, NC, USA [2]Department of Computer Science, Duke University, NC, USA [3]Computer Science & Artificial Intelligence Laboratory, Massachusetts Institute of Technology, MA, USA. Correspondence to: Jiwoo Kim <jiwoo.kim@duke.edu>, Miroslav Pajic <miroslav.pajic@duke.edu>.

*Proceedings of the 43rd International Conference on Machine Learning*, Seoul, South Korea. PMLR 306, 2026. Copyright 2026 by the author(s).

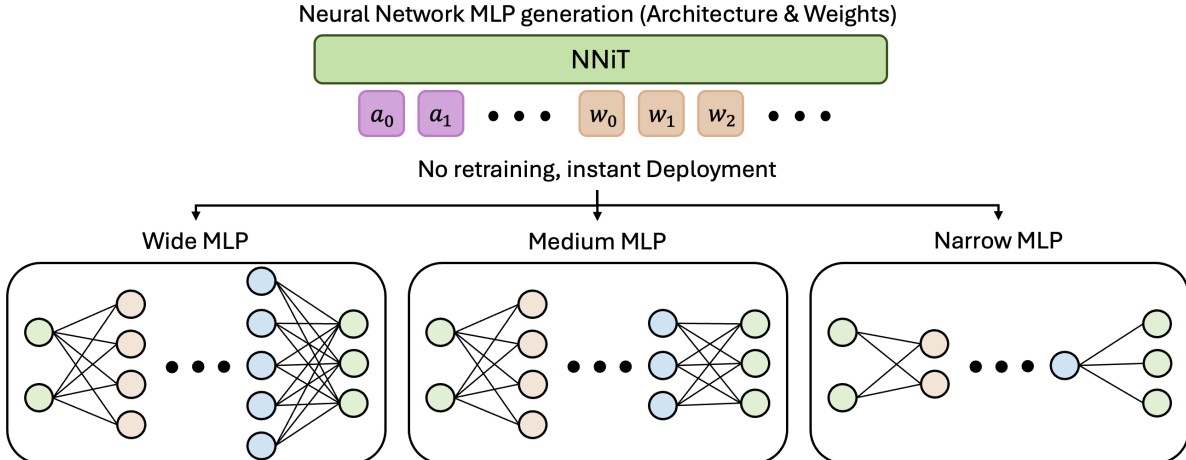

*Figure 1.* **Width-Agnostic Synthesis via Multimodal Tokenization**. Unlike existing models, the NNiT decouples functional logic from fixed matrix dimensions, allowing the zero-shot synthesis of optimal weights for architectural topologies entirely unseen during training.

solutions with arbitrary permutations resulting in unaligned parameter space. The GHN-induced parameter distribution thus provides the local structure needed for patch-based tokenization and width-agnostic transfer.

Building on this alignment, we introduce the Neural Network Diffusion Transformer (NNiT), illustrated in Figure 1, which formulates neural network synthesis as a single multimodal sequence modeling task. NNiT tokenizes aligned weight tensors into $p \times p$ patches, replacing global vectors with spatially correlated tokens. This representation makes generation width-agnostic—widening a layer just corresponds to generating additional patches without changing the tokenization scheme. An NNiT therefore learns a *joint distribution over architectures and parameters*, allowing generation of unseen architecture-weight pairs and the zero-shot synthesis of weights for user-specified structures.

We evaluate NNiT on ManiSkill3 (Tao et al., 2025) using Multilayer Perceptron (MLP) policies for robotic control. Robotic manipulation is a strict evaluation metric where small weight errors may cause task failures, unlike vision tasks where artifacts could be tolerated. Moreover, generating robotic policies is an underexplored space where enabling joint weight and architecture generation opens new avenues for research, such as meta-learning across tasks or optimizing networks for specific hardware constraints for sim to real deployment. Since this domain naturally favors MLPs, it serves as an ideal benchmark for evaluating structural generalization. Specifically, we show that NNiTs maintain $> 85\%$ success rates on architectural widths unseen during training, while baselines (Sch"urholt et al., 2024; Soro et al., 2024) fail. Beyond robotic control, we show that the same pipeline also generalizes to unseen architectures on MNIST classification.

To summarize, our primary contributions are as follows:

- We establish that GHNs align the weight space, reducing permutation-induced variability and enabling coordinate-based parameter field for tokenization.

- We introduce patch tokenization for weights, making generation width-agnostic and enabling zero-shot synthesis for unseen architecture topologies.

- We introduce NNiT, a multimodal diffusion transformer that jointly models architectures and weights for both joint generation and conditional weight synthesis.

## 2. Related Work

### 2.1. Generative Neural Network Weight Models

Parameter synthesis is increasingly formulated as a direct generative modeling problem, where neural weights are treated as structured data modalities (Wang et al., 2025a; Li et al., 2025b; Saragih et al., 2025; Erkoç et al., 2023). Recent approaches have successfully adapted diffusion (Ho et al., 2020; Song et al., 2022) and flow matching models (Lipman et al., 2023) to approximate these complex parameter distributions. However, a fundamental constraint in this domain is the high dimensionality and inherent permutation symmetry of weight spaces, where arbitrary neuron ordering obscures geometric structure.

To resolve this ambiguity, prior works have employed Variational Autoencoders (VAEs) to compress weights into permutation-invariant latent codes (Saragih et al., 2025; Soro et al., 2024). Alternatively, methods such as SANE (Sch"urholt et al., 2024) utilize explicit canonicalization methods to align the weights into a shared coordinate frame. In this vein, recent studies (Ainsworth et al., 2023; Rinaldi et al., 2025) validate that offline optimization can effectively collapse weight permutation symme-

tries. Notably, these approaches have demonstrated transferability across diverse architectural depths, successfully synthesizing parameters for Convolutional Neural Networks (CNNs) (Sch"urholt et al., 2024; Soro et al., 2024); yet, they typically remain constrained to fixed-width architectures.

## 2.2. Diffusion Models

Diffusion probabilistic models have established themselves as the standard backbone for generative tasks, demonstrating exceptional scalability across image (Peebles & Xie, 2022; Bao et al., 2023; Ma et al., 2024; Wang et al., 2025b; Tian et al., 2024) and video generation (Liu et al., 2025; Ruan et al., 2023; Kim et al., 2024a), as well as robotics (Kim et al., 2023; Chi et al., 2023; Ryu et al., 2023; Chang et al., 2023; Qi et al., 2025). Diffusion Transformers (DiTs) advanced this paradigm by utilizing patch-based tokenization to process data as sequences. This structural innovation facilitates multimodal training by projecting diverse data types—such as audio, video, and text—into a unified representation space (Liu et al., 2025; Ruan et al., 2023; Kim et al., 2024a; Shin et al., 2025; Kim et al., 2024b; Zhou et al., 2024; Gao et al., 2024). Most relevant to our framework, recent Flexible Diffusion Transformers (FiTs) (Lu et al., 2024; Wang et al., 2024) have introduced mechanisms to process variable-resolution images. We leverage these primitives to decouple parameter synthesis from fixed tensor dimensions, treating neural weights as width-agnostic fields.

## 3. Preliminaries and Problem Formulation

### 3.1. Permutation Symmetry

For an $L$-layer MLP, the parameters $\mathbf{w} = \{W_l, b_l\}_{l=1}^{L}$ are subject to inherent permutation symmetries (Hecht-Nielsen, 1990). The loss function is invariant under transformations by permutation matrices $P_l \in \mathbb{R}^{d_{l+1} \times d_{l+1}}$ (where $P_l^T P_l = I$) applied to the hidden units of layer $l$. For any set of permutations $\pi = \{P_l\}_1^L$, the input-output function of the network remains the same because the activations satisfy:

$$z_{l+1} = P^T \sigma(P W_l z_l + P b_l), \qquad (1)$$

where $\sigma$ is the non-linear activation function. This demonstrates that permuting the output of layer $l$ by $P$, while simultaneously applying the inverse permutation $P^T$ to the input of layer $l+1$, leaves the resulting activations functionally invariant (Ainsworth et al., 2023).

### 3.2. Graph HyperNetworks (GHNs)

A GHN (Zhang et al., 2020; Knyazev et al., 2021) acts as a deterministic meta-model $\Phi_\phi$ parameterized by $\phi$. Instead of optimizing weights directly via SGD, a GHN predicts the parameters $\mathbf{w}$ for a given neural architecture $\mathbf{a}$, represented as a computational graph of parameterized operations

$\mathcal{G} = (\mathcal{V}, \mathcal{E})$. Here, $\mathcal{V}$ represents the set of parameterized operations, and $\mathcal{E}$ represents the connectivity between them.

The process begins by initializing the node embeddings $\mathbf{H}^{(0)} = \{\mathbf{h}_v^{(0)}\}_{v \in \mathcal{V}}$ based on the properties of each node, such as the layer shape and operation type. These embeddings are updated iteratively via a Graph Neural Network (GNN). At step $t$, the embedding for node $v$ is updated aggregating messages from these neighbors:

$$\mathbf{h}_v^{(t+1)} = U_\theta \left( \mathbf{h}_v^{(t)}, \sum_{u \in \mathcal{N}(v)} M_\psi(\mathbf{h}_u^{(t)}) \right), \qquad (2)$$

where $M_\psi(\cdot)$ is a message function that projects neighbor states into the message space, $U_\theta(\cdot)$ is a node update function that integrates the aggregated neighborhood information with the node's current state, and $\mathcal{N}(v)$ denotes the set of incoming neighbors for node $v$. After T steps of propagation, a shared decoder network $D_\phi$ projects the final node embedding into the weight parameters for that specific layer $\mathbf{w}_v = D_\phi(\mathbf{h}_v^{(T)})$.

### 3.3. Problem Objective

We formulate our generative model as learning a joint distribution $p_\theta(\mathbf{a}, \mathbf{w})$ over the space of topologies $\mathbf{a} \in \mathcal{A}$ and parameters $\mathbf{w} \in \mathcal{W}$. A functional network is defined by the tuple $\tau = (\mathbf{a}, \mathbf{w})$.

**Objective.** Our primary goal is to achieve width-agnostic neural network MLP generation, enabling the model to generalize to unseen topologies not present in the training set. Existing generative approaches typically flatten weights into fixed-dimensional vectors, rigidly coupling the generative prior to specific architectures, or similar architectures with different depths (Sch"urholt et al., 2024; Soro et al., 2024).

**Problem.** Designing methods for such width-agnostic synthesis is fundamentally difficult due to the permutation symmetry and lack of geometry described in Section 3.1. In standard neural networks trained via SGD, adjacent weights in a matrix are uncorrelated due to the arbitrary ordering of neurons, breaking the core structure required for spatial generation.

**Key Insight and Proposed Approach.** To address this challenge, we demonstrate that GHNs with a CNN decoder inherently generate a structurally aligned weight space. We observe that, unlike SGD, which produces unstructured weight matrices, GHNs collapse permutation symmetries into a consistent topological structure. Leveraging this finding, we propose *treating neural network weights not as independent vectors, but as a continuous spatial field*. This structural alignment allows us to employ image-based generative backbones to model the conditional likelihood $p_\theta(\mathbf{w}|\mathbf{a})$ across varying topologies as well as joint multimodal generation of $p_\theta(\mathbf{a}, \mathbf{w})$.

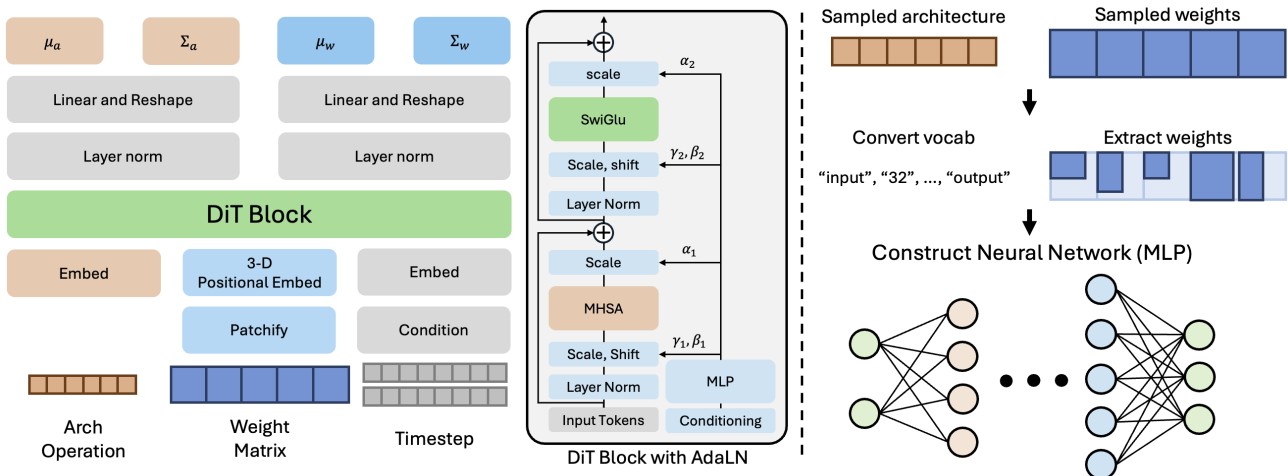

*Figure 2.* **NNiT Framework Overview. Left: Unified Generative Architecture.** We formulate neural synthesis as a multimodal sequence task. Discrete architecture tokens (orange) and continuous weight matrices (blue) are unified into a single sequence, with weights processed as spatially correlated patches. A Diffusion Transformer (DiT) models the joint distribution using per-modality timestep conditioning ($\mu_a, \Sigma_a, \mu_w, \Sigma_w$) via the Mixture of Noise Levels (MoNL) framework, enabling both co-design $p(\mathbf{a}, \mathbf{w})$ and conditional synthesis $p(\mathbf{w}|\mathbf{a})$. **Right: Deployment Pipeline.** During inference, sampled architecture tokens are decoded into layer widths. The generated weight tensors are then extracted to match these target dimensions, assembling a directly executable MLP.

# 4. Structural Alignment via Graph-Conditioned CNN Decoder

Our NNiT framework builds on the empirical observation that GHNs, when paired with a CNN decoder, produce weight tensors that are consistently organized and exhibit reliable local correlations. This effectively narrows the training distribution seen by our downstream generative model; instead of learning over the full, permutation-ambiguous space of MLP parameters, it learns over a more structurally coherent subset of MLPs induced by the GHN generator. This addresses the two main obstacles to field-based weight generation: permutation ambiguity and the lack of stable spatial structure.

We attribute the reduction in permutation-induced variability to the GHN's parameterization mechanism. The GHN computes embeddings $\mathbf{h}_v$ for each node by propagating information over the architecture graph. The input node $\mathcal{V}_{in}$ is anchored by task-specific features, providing a consistent reference across samples. A shared decoder $D_\phi(\cdot)$ then maps these embeddings to the weight tensors of each layer.

Crucially, we implement $D_\phi$ as a CNN decoder. The CNN decoder maps a compact node embedding to a full parameter tensor, $D_\phi : \mathbb{R}^d \to \mathbb{R}^N$ with $d \ll N$. It introduces an explicit locality bias on the generated weights where nearby indices are produced from shared latent features. This yields repeatable spatial patterns like the vertical banding structures seen in Figure 3. We also see consistent local correlations across seeds.

In contrast, SGD-trained MLPs have parameterizations with

no stable alignment. This GHN generator thus produces a weight distribution with reliable local structure, which is the prerequisite needed for patch-based tokenization and width-agnostic generation using NNiT.

# 5. Neural Network Diffusion Transformers

Leveraging the structurally aligned weight-space induced by the GHN, we introduce the NNiT, a multimodal framework that unifies neural architecture search and parameter generation into a single sequence modeling task (Figure 2).

## 5.1. Unified Neural Network Representation

We define a joint embedding space that processes discrete architecture tokens and continuous weight patches simultaneously.

### 5.1.1. ARCHITECTURE AS DISCRETE TOKENS

We formulate the architecture $\mathbf{a} \in \mathcal{V}^S$ as a sequence of discrete tokens, where $\mathcal{V}$ is the vocabulary of layer widths and $S$ denotes the network depth. In contrast to approaches that utilize adjacency matrices to model arbitrary branching topologies (An et al., 2023), we formulate the architecture as a discrete sequence. This representation is sufficient for MLP policies where full layer connectivity is implicit in the sequence order. Accordingly, each token $a_i$ serves as a discrete index mapping to a specific neuron count within $\mathcal{V}$. These indices are projected to the model's hidden dimension $d$ via a learnable embedding table $E_a$, resulting in the dense vector sequence $\mathbf{z}_a \in \mathbb{R}^{S \times d}$.

## 5.1.2. WEIGHTS AS PATCHIFIED CONTINUOUS TENSOR

To achieve width-agnostic generation, we leverage the structural alignment of the structurally aligned weight spaces to frame the weights as a continuous 4D tensor. For a specific layer $l$, we concatenate the weight matrix and bias vector to form a parameter block $B_l \in \mathbb{R}^{n_l \times (n_{l+1}+1)}$. To accommodate diverse topologies within a single generative model, we standardize these blocks by padding them onto a maximal coordinate grid $H \times W$, where $H = M$, $W = M + p$, with $M$ being an arbitrary maximum supported width following (Lu et al., 2024). The complete set of network parameters is thus represented as a 4D tensor $\tilde{\mathbf{w}} \in \mathbb{R}^{(S-1) \times 1 \times H \times W}$.

We then decompose this tensor into non-overlapping $p \times p$ patches locally within each layer. This results in a sequence of flattened patches $\mathbf{w}_{patch} \in \mathbb{R}^{(S-1) \cdot N \times p^2}$, where $N = \frac{H \cdot W}{p^2}$ is the number of patches per layer grid. These patches are linearly projected to the model dimension $d$ to form the continuous embedding sequence $\mathbf{z}_w \in \mathbb{R}^{L_{seq} \times d}$.

The final input to the transformer is the unified sequence $\mathbf{z} = [\mathbf{z}_a; \mathbf{z}_w]$, allowing the self-attention mechanism to model the joint dependencies between the discrete topology tokens and the functional weight patches.

## 5.2. Transformer Backbone

NNiT processes the unified sequence $\mathbf{z}$ using a standard Diffusion Transformer (DiT) backbone (Peebles & Xie, 2022). Each block consists of multi-head self-attention and a feed-forward network utilizing SwiGLU activations like LLaMA (Touvron et al., 2023).

To handle the heterogeneous nature of the input, conditioning is implemented via Adaptive Layer Norm (AdaLN-Zero) with dual-timestep embeddings. We employ distinct timestep embedding models for the architecture ($t_a$) and weights ($t_w$). These embeddings are concatenated to dynamically regulate the scale and shift parameters of the layers, allowing the network to modulate its processing based on the noise level of each modality.

## 5.3. Training with Mixture of Noise Levels (MoNL)

Finally, we adopt the Mixture of Noise Levels (MoNL) framework (Kim et al., 2024a) to unify architecture generation and parameter synthesis; we provide more details in Appendix A. During training, we stochastically sample between two noise scheduling modes for each batch:

- Joint Generation Mode ($t_a = t_w > 0$): Both architecture and weights are diffused to the same timestep $t_{ref}$. The model learns the joint distribution $p(\mathbf{a}, \mathbf{w})$, enabling it to propose novel architecture-weight pairs from scratch.

- Conditional Synthesis Mode ($t_a = 0, t_w > 0$): Architecture tokens remain noise-free ($t_a = 0$) while weights are diffused to $t_{ref}$. This forces the model to learn the conditional distribution $p(\mathbf{w}_{t-1}|\mathbf{w}_t, \mathbf{a}_0)$, effectively teaching it to synthesize valid weights for a fixed topology.

To optimize this joint distribution, we minimize a composite loss function. Given clean data $(\mathbf{a}_0, \mathbf{w}_0)$ and independent Gaussian noise $\epsilon_a, \epsilon_w \sim \mathcal{N}(0, \mathbf{I})$, the total objective combines the standard noise prediction error with a variational lower bound term $\mathcal{L}_{vb}$ to learn the covariance $\Sigma_\theta$ – i.e.,

$$\mathcal{L}_{total} = \mathbb{E}_{t_a, t_w, \epsilon} \left[ \|\epsilon_\theta^a - \epsilon_a\|^2 + \|\epsilon_\theta^w - \epsilon_w\|^2 \right] + \mathcal{L}_{vb}, \quad (3)$$

where $\epsilon_\theta^a$ and $\epsilon_\theta^w$ denote the model's noise predictions for the architecture and weight modalities, respectively.

## 5.4. Deployment and Synthesis

Inference adapts to the presence of architectural constraints. If a target architecture is provided, the model samples $\mathbf{w} \sim p(\mathbf{w}|\mathbf{a})$. Otherwise, it samples from the joint distribution $(\mathbf{a}, \mathbf{w}) \sim p(\mathbf{a}, \mathbf{w})$.

The final network assembly is a deterministic projection driven by the architecture tokens. The sequence $\hat{\mathbf{a}}$ is decoded into integer widths $[n_1, \ldots, n_S]$ where $n_1, ..., n_S \in V$. These dimensions serve as a cropping mask for the generated weight tensor. We extract the valid submatrices $\mathbf{w}_l \in \mathbb{R}^{n_l \times (n_{l+1}+1)}$ from the maximal grid and discard the padding.

## 6. Experimental Evaluation

We aim to demonstrate that the NNiT learns a width-agnostic representation of neural weights, enabling transferability across diverse architectures. In particular, we consider the following research questions:

- **Does the GHN backbone effectively reduce permutation ambiguity?** We investigate if GHNs induce the necessary structural alignment to treat weights as local patches (Section 6.1).

- **Is the NNiT more effective than baselines in zero-shot transfer?** We evaluate whether our patch-based tokenization outperforms existing baseline models (SANE (Sch"urholt et al., 2024) and D2NWG (Soro et al., 2024)) when synthesizing weights for architectural topologies with diverse widths (Section 6.4).

- **Can NNiT perform multimodal joint generation?** We assess the NNiT's ability to model the joint distribution $p(\mathbf{a}, \mathbf{w})$, spontaneously generating both diverse network topologies and their functional parameters (Section 6.5).

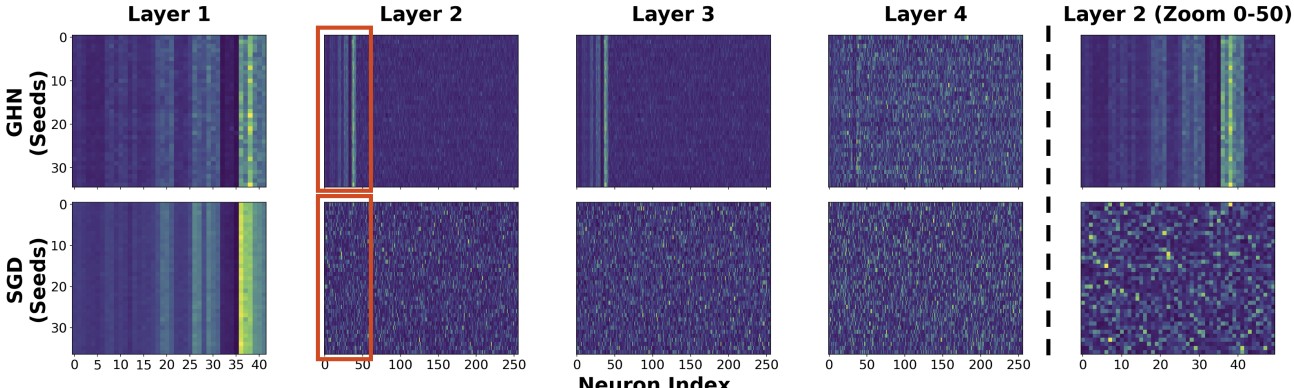

*Figure 3.* **Visualizing Structural Alignment and Induced Geometry.** Comparison of the weight magnitude profiles across 35 independent seeds. **Top (GHN):** The consistent alignment across the seeds demonstrates that the GHN **1) successfully spatially aligns the weight spaces**, effectively resolving the permutation ambiguity inherent in neural networks. Moreover, the visible structural banding indicates that the GHN **2) imposes meaningful geometric structure** (spatial correlation), transforming independent parameters into a spatially aligned space. **Bottom (SGD):** In contrast, SGD weights exhibit unstructured noise due to arbitrary permutations, lacking both structural alignment and local spatial geometry.

*Table 1.* **Alignment vs. Performance.** Evaluation of 35 independent seeds. Both methods achieve optimal performance, yet GHNs induce the structural alignment necessary for tokenization without sacrificing weight diversity, confirming the absence of mode collapse.

| Metric | GHN | SGD |
|---|---|---|
| *Policy Performance* | | |
| Mean Return | 39.35 | 39.49 |
| Mean Success Rate | 99.0% | 98.9% |
| *Weight Diversity (Nearest Neighbor)* | | |
| $L_2$ Distance ($\uparrow$) | 54.66 | 41.73 |
| Cosine Similarity ($\downarrow$) | 0.07 | 0.02 |

### 6.1. Empirical Validation of Structural Alignment

We first test the hypothesis that GHNs induce the structural conditions necessary for patch-based generation. Using `PickCube-v1` as a representative environment, we compare weight-magnitude profiles from 35 independently sampled GHN-generated policies and 35 standard SGD-trained networks. Both sets use the same MLP architecture, $[\text{input}, 256, 256, 256, \text{output}]$.

As shown in Table 1, both methods achieve expert performance ($> 99\%$ success). Figure 3 nevertheless shows a clear structural difference. Because the input feature space is fixed across seeds, both GHN and SGD exhibit pronounced vertical banding in the input projection layer. Beyond the first layer, however, SGD-trained weights display little consistent spatial structure across seeds.

On the other hand, GHN-generated weight-magnitude heatmaps retain vertically banded, locally correlated structure across all layers. This suggests that generating parameters through the shared, graph-conditioned decoder selects MLPs with aligned weight spaces.

Crucially, this alignment is not a result of mode collapse. The pairwise nearest neighbor analysis in Table 1 reveals that GHN policies maintain high weight diversity with a mean $L_2$ distance of 54.66 and a small mean cosine similarity of 0.07. Additional per-architecture heatmaps and diversity metrics in Appendix B further support this finding. We also provide additional evidence in Appendix C, where a centered kernel alignment (CKA) analysis confirms the absence of mode collapse and a permutation ablation establishes that the NNiT depends on this alignment.

### 6.2. Dataset & Experimental Design

To evaluate the neural network generation, we constructed a dataset of 4-hidden-layer MLP policies with layer widths sampled from $\mathcal{V} \in \{\text{input}, 16, 32, 64, \text{output}\}$. We partitioned 72 distinct topological configurations into 64 training architectures and 8 held-out configurations to test zero-shot generalization to unseen structures. We trained 128 independent GHNs, extracting the top-100 performing weight configurations per training architecture. This yielded a final corpus of $6,400$ expert policies for training our NNiT model. Additional details are provided in Appendix D, with full training hyperparameters summarized in Appendix E.

### 6.3. Evaluation Protocol

We assess the generation quality by directly deploying synthesized policies into the ManiSkill3 environment, evaluating each over 50 episodes to measure control fidelity. Following the standard protocols (Liang et al., 2024; Sch"urholt et al., 2024; Soro et al., 2024), we report the performance

*Table 2.* **Architecture-conditioned weight generation** ($p(\mathbf{w}|\mathbf{a})$). We compare NNiT against baseline methods on both seen and **unseen (zero-shot)** architectural configurations. Reporting the top-10 performance metric, we observe that while the gap is negligible on training configurations, performance diverges sharply on unseen topologies. Baselines degrade significantly, while NNiT maintains robust performance, validating its effectiveness as a width-agnostic neural network generator.

| | Architecture | PickCube-v1 | | PushCube-v1 | | StackCubeEasy-v1 | |
|---|---|---|---|---|---|---|---|
| | | Return | Success | Return | Success | Return | Success |
| Dataset | | $39.37 \pm 0.55$ | $98\%$ | $42.02 \pm 0.4$ | $100\%$ | $39.53 \pm 0.78$ | $94\%$ |
| SANE | | $3.81 \pm 0.29$ | $1\%$ | $6.90 \pm 0.25$ | $6\%$ | $4.64 \pm 0.16$ | $0\%$ |
| D2NWG | Seen | $38.59 \pm 1.35$ | $98\%$ | $42.28 \pm 0.05$ | $100\%$ | $37.65 \pm 0.69$ | $90\%$ |
| **NNiT (Ours)** | | $38.77 \pm 0.41$ | $98\%$ | $42.10 \pm 0.16$ | $100\%$ | $38.38 \pm 0.51$ | $91\%$ |
| SANE | | $3.65 \pm 0.41$ | $1\%$ | $5.89 \pm 0.32$ | $2\%$ | $3.71 \pm 0.13$ | $0\%$ |
| D2NWG | **Unseen** | $28.31 \pm 4.53$ | $59\%$ | $37.35 \pm 1.63$ | $89\%$ | $27.18 \pm 2.35$ | $42\%$ |
| **NNiT (Ours)** | | $\mathbf{38.70 \pm 0.36}$ | $\mathbf{99\%}$ | $\mathbf{42.00 \pm 0.19}$ | $\mathbf{100\%}$ | $\mathbf{36.24 \pm 1.33}$ | $\mathbf{86\%}$ |

*Table 3.* **Multimodal Joint Synthesis** ($p(\mathbf{a}, \mathbf{w})$). We evaluate NNiT's capacity to spontaneously synthesize complete functional policies. As no baselines support joint synthesis, NNiT is presented in isolation. The results demonstrate near-perfect success rates ($99\%$–$100\%$) across PickCube and PushCube and $90\%$ for StackCubeEasy. This confirms that the model can generate high-performing policies without a fixed architectural prompt.

| | PickCube-v1 | | PushCube-v1 | | StackCubeEasy-v1 | |
|---|---|---|---|---|---|---|
| | Return | Success | Return | Success | Return | Success |
| NNiT(Ours) | $38.85 \pm 0.32$ | $99\%$ | $41.99 \pm 0.10$ | $100\%$ | $37.57 \pm 0.85$ | $90\%$ |

of the top-10 policies selected from 100 generated samples, simulating an offline validation phase before deployment. To benchmark width-agnostic synthesis methods, we compare NNiT against D2NWG (Soro et al., 2024) and SANE (Sch"urholt et al., 2024), trained on the same expert dataset. As no prior methods target zero-shot architectural generalization, we adapted both baselines to this setting and evaluated them without test-time optimization.

### 6.4. Zero-Shot Width Transferability

Table 2 presents the definitive test of our framework regarding zero-shot conditional generation. We evaluate the model's ability to synthesize weights for architectures with different topologies (i.e., $p(\mathbf{w}|\mathbf{a})$).

On seen architectures, both the NNiT and the D2NWG (Soro et al., 2024) baseline achieve near perfect performance for all three tasks. These results suggest that the D2NWG (Soro et al., 2024) effectively captures the training distribution via explicit architecture conditioning. We also note that part of the D2NWG's strong performance on seen data is due to the GHN pre-aligning the MLP weight spaces, making them more conducive to downstream weight-space learning.

However, such conditioning rigidly couples the generation process to specific topologies because the D2NWG relies on fixed zero-padded vectorization schemes lacking explicit architectural awareness. This limitation impairs

transferability to unseen structures and causes success rates on `PickCube-v1` and `StackCubeEasy-v1` to drop to $59\%$ and $42\%$. While the D2NWG retains $89\%$ success on `PushCube-v1`, we attribute this to task simplicity, tolerating higher precision errors rather than true structural generalization.

In contrast, SANE (Sch"urholt et al., 2024) fails to produce functional policies across both seen and unseen topologies. We attribute this failure to the reliance on global positional embeddings for the sequence order encoding, as this formulation becomes unstable when trained on a mixture of diverse network widths. Because SANE creates tokens by slicing layer weights row-wise, any variation in the layer width alters the token count and shifts the global indices for all subsequent layers. This index shift misaligns learned positional semantics and prevents the model from constructing coherent policies even within the training distribution.

On the other hand, NNiT performs very close to the training set with over $85\%$ success across all tasks. We attribute this robustness specifically to our patch-based tokenization. By decomposing weights into locally consistent patches, NNiT decouples the generative prior from the global weight space dimensions. Synthesizing a wider layer is analogous to increasing the resolution of a generated image, where the model aggregates a larger number of valid functional patches. The same idea extends naturally to depth—adding layers corresponds to appending additional architecture to-

*Table 4.* **Architecture Diversity & Generalization.** Specific functional networks sampled from the generated distribution. Note that policy architectures list hidden layer widths only, omitting input and output layers. The bottom section highlights Zero-Shot Synthesis on topologies entirely unseen (Test Set) during training. Notably, NNiT achieves 98% success on the unseen configuration [input, 32, 16, 16, 16, output].

| Policy Architecture | Partition | Return | Success |
|---|---|---|---|
| *Training (Seen) Topologies* | | | |
| 32, 16, 64, 32 | Seen | 39.69 | 100% |
| 32, 64, 16, 32 | Seen | 39.01 | 100% |
| 32, 32, 32, 16 | Seen | 38.99 | 100% |
| 64, 64, 16, 32 | Seen | 38.81 | 98% |
| *Zero-Shot (Unseen) Topologies* | | | |
| 32, 32, 16, 64 | Unseen | 37.75 | 96% |
| 32, 32, 32, 64 | Unseen | 37.01 | 94% |
| 32, 16, 16, 16 | Unseen | 38.53 | 98% |
| 16, 16, 16, 32 | Unseen | 33.80 | 94% |

kens and their associated weight blocks to the sequence, mirroring the scalability of joint audio-video generation.

We report the full architecture performance distribution across all 80 generated policies in Appendix F and provide further analysis on a single topology in Appendix G. We further verify that the pipeline transfers beyond robotic control by evaluating on MNIST classification (results summarized in Appendix H); we show that the NNiT attains 96.1% mean test accuracy across unseen architectures.

### 6.5. Multimodal Joint Synthesis

A unique advantage of our patch-based representation is that it unifies the typically disjoint problems of Neural Architecture Generation (NAG) (An et al., 2023) and parameter generation into a single sequence modeling task. By interleaving discrete architecture tokens with continuous weight patches, NNiT learns the joint distribution $p(\mathbf{a}, \mathbf{w})$, allowing the discrete topology to serve as a causal prompt that dynamically modulates weight generation.

The results in Table 3 quantify this capability. NNiT spontaneously generates complete network policies that achieve near-perfect success rates (99%–100%) across tasks. This confirms that the NNiT effectively captures the conditional dependencies between topological bottlenecks and functional parameters.

Crucially, the model does not collapse to memorizing training templates. As summarized in Table 4 (Bottom), NNiTs generate functional policies for unseen configurations, such as [input, 32, 16, 16, 16, output]. The high success rate (98%) on these unseen policies validates that the NNiTs have internalized the structural logic of neural design,

extrapolating to the broader search space rather than simply recalling memorized instances. As such, we can spontaneously generate diverse, high-performing neural network policies. We report the full distribution of the generated policies in Appendix I and provide visual rollouts in Appendix J.

## 7. Limitations and Future Work

Our experiments currently cover MLPs with up to four hidden layers due to computational limits. This is not a limitation of NNiT—patch tokenization supports larger widths and depths, but deeper models are harder to train as neural weights are continuous and unbounded, with high variance. Future work will adapt stabilization techniques from pixel space diffusion models (Yu et al., 2025) to better handle this dynamic range.

Structurally, framing the network depth as a temporal dimension and weight matrices as spatial features renders our approach analogous to video synthesis. This alignment allows NNiT to leverage efficiency optimizations from Video Diffusion Transformers (Chen et al., 2025), such as linear attention mechanisms (Xie et al., 2024). These techniques reduce quadratic complexity, facilitating the training of general-purpose generators capable of synthesizing billion-parameter foundation models.

Finally, NNiTs support flexible deployment for embodied AI. Rather than training separate policies for each target architecture, a single generator can be used to synthesize weights that satisfy user-specified constraints, such as a width or a compute budget. Moreover, because NNiT conditions weight generation on discrete tokens, the same mechanism could be extended beyond architecture to other conditioning signals—such as environment or task tokens—enabling meta-learning and rapid adaptation to changing configurations.

## 8. Conclusion

In this work, we set out to address two barriers to generating fully-functional neural networks: the coupling of weight generation and fixed matrix dimensions, and the permutation symmetries that make MLP parameterizations inherently unaligned. The Neural Network Diffusion Transformers (NNiTs) resolve the first by replacing global vectorization with patch representation, so widening a layer corresponds to generating additional tokens rather than changing the token space. We resolve the second barrier by leveraging Graph HyperNetworks (GHNs) with a CNN decoder, which induces consistently organized weight tensors with reliable local correlations, making patch-based modeling feasible.

With this structure in place, NNiTs treat discrete architec-

ture tokens and continuous weight patches as a single multimodal sequence, enabling both joint architecture-weight generation and conditional weight synthesis. We demonstrate that on ManiSkill3, this design yields robust zero-shot generalization to architectures and widths unseen during training, while vectorized baselines fail.

## Acknowledgements

This work is sponsored in part by the AFOSR under the award number FA9550-19-1-0169, and by the NSF under NAIAD Award 2332744 as well as the National AI Institute for Edge Computing Leveraging Next Generation Wireless Networks, Grant CNS-2112562.

## Impact Statement

This paper advances the methodology of neural network generation and weight space representation learning, contributing to the broader field of Machine Learning. There are many potential societal consequences of our work, none of which we feel must be specifically highlighted here.

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

## A. Mixture of Noise Level (MoNL) Diffusion Framework

We adapt the MoNL framework to a dual-modality state space consisting of discrete architectural tokens and continuous weight parameters. We employ a simplified subset of the original mixing strategies, utilizing vanilla joint diffusion and per modality (Pm). This enables simultaneous modeling of the joint synthesis distribution and the use of architecture as a clean conditional prior.

Let the state be denoted by $\mathbf{z}_0 = [\mathbf{z}_0^{(a)}; \mathbf{z}_0^{(w)}]$ where $\mathbf{z}_0^{(a)} \in \mathbb{R}^{S \times d}$ represents the embedded architecture tokens and $\mathbf{z}_0^{(w)} \in \mathbb{R}^{L_{seq} \times d}$ represents the projected weight patches. We employ a vectorized timestep $\mathbf{t} = (t_a, t_w)$ to independently control the noise level of each modality.

To train the model for both joint design and conditional weight generation, we sample $\mathbf{t}$ via a reference timestep $t_{\text{ref}} \sim \mathcal{U}(1, T)$ and a mode indicator $m \sim \mathcal{U}(0, 1)$. The effective timesteps are assigned as follows:

$$t_w = t_{\text{ref}}, \quad t_a = \begin{cases} t_{\text{ref}} & \text{if } m \geq 0.5 \text{ (Joint Training)}, \\ 0 & \text{if } m < 0.5 \text{ (Conditional Training)}. \end{cases} \tag{4}$$

When $m < 0.5$, we enforce $t_a = 0$, effectively conditioning the weight generation on a clean architecture embedding. To enforce the clean boundary condition, we define a target noise tensor $\tilde{\boldsymbol{\epsilon}} = [\tilde{\boldsymbol{\epsilon}}^{(a)}; \tilde{\boldsymbol{\epsilon}}^{(w)}]$ where $\tilde{\boldsymbol{\epsilon}}^{(k)} = \boldsymbol{\epsilon} \sim \mathcal{N}(0, \mathbf{I})$ if $t_k > 0$ and $\tilde{\boldsymbol{\epsilon}}^{(k)} = \mathbf{0}$ if $t_k = 0$. We train a joint denoising network $\epsilon_\theta(x_\mathbf{t}, \mathbf{t})$ to predict this target noise while simultaneously learning the variance $\Sigma_\theta$. The total training objective combines Mean Squared Error (MSE) with the variational lower bound ($\mathcal{L}_{\text{vb}}$):

$$\mathcal{L}_{\text{total}} = \sum_{k \in \{a, w\}} \left( \underbrace{\|\tilde{\boldsymbol{\epsilon}}^{(k)} - \epsilon_\theta^{(k)}(x_\mathbf{t}, \mathbf{t})\|^2}_{\text{MSE}} + \mathcal{L}_{\text{vb}}^{(k)} \right). \tag{5}$$

For the variational term, we utilize discretized Gaussian log-likelihoods for the architecture embeddings to handle their discrete structure, and standard Continuous Gaussian Log-Likelihoods for the weight parameters.

## B. Dataset Distribution Analysis

We provide supporting visualizations and quantitative metrics to validate the structural properties proposed in Section 6.1.

Figure 5 provides empirical evidence that the GHN's explicit locality bias projects parameters onto a structurally aligned field. The distinct vertical banding patterns observed across the 100 filtered weight samples indicate that specific spatial regions consistently encode identical functional roles across independent generations. We further note that tasks sharing identical state spaces (`PickCube-v1` and `StackCubeEasy-v1`) exhibit similar alignment signatures. This confirms that the GHN effectively propagates the fixed topological anchors of the input and output layers through the hidden topology.

We analyze the distributional properties of the weights in Table 5 to verify that this structural alignment does not induce mode collapse. We compute the mean Nearest Neighbor (NN) metrics for the 100 independent samples of a fixed architecture. The results show high mean Euclidean distances (46.94–72.79) and low cosine similarities (0.12–0.21) across all evaluated tasks. Figure 4 complements these metrics, illustrating that the dataset retains significant parametric dispersion.

## C. Analyses of Structural Alignment

GHN-generated weights exhibit structural alignment across seeds, whereas SGD-trained networks do not (Section 6.1). We investigate this alignment with two analyses. A permutation ablation destroys the structural alignment while preserving all

*Table 5.* **Quantitative Diversity Metrics.** Evaluation of 100 filtered policy weights. The high $L_2$ distances and low cosine similarities confirm that the dataset maintains significant parametric diversity despite the structural alignment.

| Environment | Mean NN $L_2$ ($\uparrow$) | Mean NN Cosine ($\downarrow$) |
|---|---|---|
| PickCube-v1 | 46.94 | 0.17 |
| PushCube-v1 | 49.66 | 0.12 |
| StackCubeEasy-v1 | 72.79 | 0.21 |

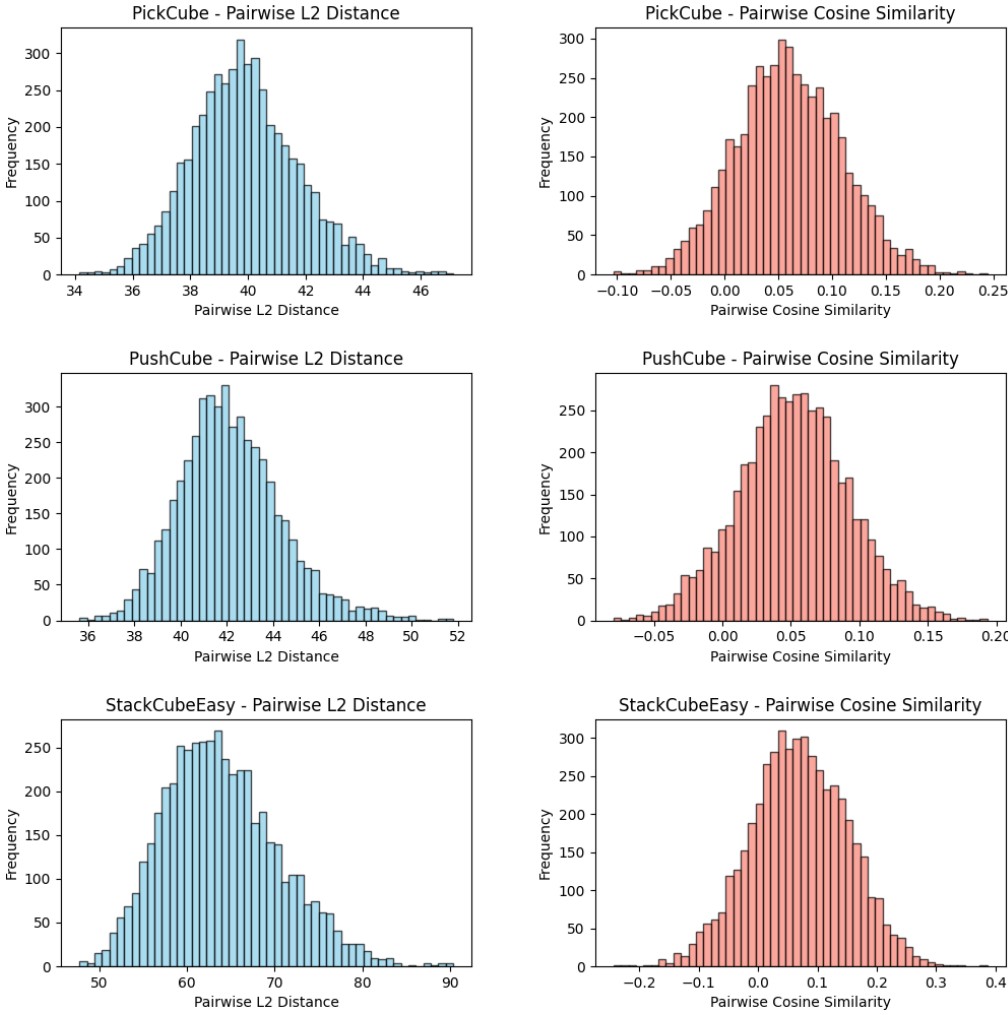

*Figure 4.* **Dataset Diversity Analysis.** Histograms of pairwise $L_2$ distances and Cosine Similarities across all three environments. The consistently high $L_2$ distances and low cosine similarities confirm that the structural alignment imposed by the GHN does not result in mode collapse.

*Table 6.* **Representational Diversity via CKA.** Lower CKA indicates greater functional diversity between independently trained networks. GHN-generated networks exhibit lower CKA than SGD-trained networks at every layer, with the gap widening in deeper layers.

| CKA | GHN | SGD |
|---|---|---|
| Overall | $0.849 \pm 0.088$ | $0.933 \pm 0.030$ |
| Layer 1 | $0.917 \pm 0.016$ | $0.957 \pm 0.005$ |
| Layer 2 | $0.880 \pm 0.033$ | $0.926 \pm 0.022$ |
| Layer 3 | $0.751 \pm 0.083$ | $0.918 \pm 0.037$ |

weight statistics, showing that NNiT's width-agnostic generation depends on the alignment. A centered kernel alignment (CKA) analysis then confirms that the aligned weights remain functionally diverse, ruling out mode collapse.

### C.1. Representational Similarity Analysis via CKA

We compute pairwise CKA between the GHN-generated and SGD-trained $[256, 256, 256]$ MLPs on `PickCube-v1`. Lower CKA indicates greater functional diversity, so a collapsing generator would manifest as *higher* CKA among GHN-generated networks.

As reported in Table 6, GHN-generated networks exhibit lower CKA than SGD-trained networks at every layer, with the gap

*Table 7.* Hyperparameters for Dataset Generation and Filtering

| Parameter | Environment Specifics | | |
| --- | --- | --- | --- |
| | PickCube-v1 | PushCube-v1 | StackCubeEasy-v1 |
| *GHN Training* | | | |
| Iterations | 500 | 500 | 1500 |
| Beta Decay | 0.97 | 0.97 | 0.98 |
| Buffer Size | 1000000 | 1000000 | 1000000 |
| Meta Batch Size | 8 | 8 | 16 |
| Start LR | 1e-3 | 1e-3 | 1e-3 |
| End LR | 7e-4 | 7e-4 | 7e-4 |
| *Filtering Criteria* | | | |
| Min Success Rate ($\tau_{success}$) | 0.9 | 0.9 | 0.8 |
| Min Return ($\tau_{return}$) | 35.0 | 35.0 | 30.0 |
| Policies / Arch ($N_{policy}$) | 100 | 100 | 100 |

widening in deeper layers (Layer 3: 0.751 vs. 0.918). Under mode collapse, the opposite would hold. Together with the weight-diversity metrics in Table 1, this confirms that the GHN supplies spatial alignment without collapsing functional diversity.

### C.2. Random Permutation Ablation

We apply random neuron permutations to every GHN-generated MLP policy in the training corpus. This preserves the input-output function and all per-layer weight statistics, but scrambles the row-column ordering. Training NNiT on the permuted corpus collapses zero-shot performance from $78.8\%$ to $0\%$ (0/80 policies) on PickCube-v1, by the same mechanism that breaks SANE (Sch"urholt et al., 2024).

The permutation also yields a controlled approximation of SGD-style weight populations. Intact GHN-generated weights exhibit a cross-seed magnitude correlation of $r \approx 0.65$, whereas both permuted GHN-generated and SGD-trained weights drop to $r \approx 0.00$. Because the permutation leaves the per-layer weight statistics unchanged, the drop reflects the loss of spatial structure rather than a change in the weight distribution.

## D. Dataset Generation Methodology

We construct a dataset of neural network policies using a teacher-student distillation pipeline leveraging the MLP Graph HyperNetworks (GHNs) from HyperPPO (Hegde et al., 2023).

We target three robotic manipulation tasks from the ManiSkill3 benchmark (Tao et al., 2025): PickCube-v1, PushCube-v1, and a custom variant StackCubeEasy-v1. We designed StackCubeEasy-v1 to ensure target stability by instantiating the goal object as a kinematic body to remain stationary during interaction.

The ground-truth behavioral prior is derived from a single multi-task expert policy trained via Proximal Policy Optimization (PPO). This expert utilizes a 3-layer Multi-Layer Perceptron (MLP) architecture with hidden dimensions of $[256, 256, 256]$ and Tanh activations. The expert is trained for 10M to 50M steps across parallel environments using the standard RL baseline procedure.

To populate the architectural search space, we train an ensemble of 128 independent GHNs from scratch ($H_\phi$), initialized with distinct random seeds. This procedure ensures the final corpus captures the parametric variance inherent to the GHN optimization landscape.

We populate the final dataset by sampling from a discrete search space of 4-layer MLPs. Layer widths are drawn from the vocabulary $\{input, 16, 32, 64, output\}$. Architectures with a width of 16 at depths 3 and 4 are excluded due to insufficient capacity. This process yields a training set of 64 distinct architectures and a held-out test set of 8 unseen topologies. For each candidate, we generate weights and evaluate policy performance over $N_{eval} = 64$ episodes. We retain the top-100 validated policies per architecture meeting task-specific success thresholds, resulting in a corpus of 6,400 expert policies (Table 7). All weights are globally normalized to a target standard deviation to align the parameter distribution for diffusion training (Peebles et al., 2022).

*Table 8.* **Hyperparameter Configuration.** Training and optimization settings for the diffusion process.

| Hyperparameter | PickCube | PushCube | StackCubeEasy |
|---|---|---|---|
| *Diffusion Process* | | | |
| Timesteps ($T$) | 1000 | 1000 | 1000 |
| Noise Schedule | Linear | Linear | Linear |
| Prediction Target | Noise ($\epsilon$) | Noise ($\epsilon$) | Noise ($\epsilon$) |
| *Optimization* | | | |
| Optimizer | AdamW | AdamW | AdamW |
| Weight Decay | 0.0 | 0.0 | 0.0 |
| Grad Clip | 1.0 | 1.0 | 1.0 |
| EMA Rate | 0.9999 | 0.9999 | 0.9999 |
| Precision | bfloat16 | bfloat16 | bfloat16 |
| LR Schedule | Cosine | Cosine | Cosine |
| Warmup Steps | 500 | 500 | 500 |
| Peak LR | $7 \times 10^{-5}$ | $7 \times 10^{-5}$ | $7 \times 10^{-5}$ |
| Min LR | $3 \times 10^{-5}$ | $3 \times 10^{-5}$ | $3 \times 10^{-5}$ |
| Batch Size | 16 | 16 | 16 |
| Training Epochs | 1300 | 1000 | 1300 |

*Table 9.* **Full Distribution on Unseen Architectures.** Mean (over all 80 generated policies), Top-10, and Worst-Arch. NNiT's worst single architecture outperforms D2NWG's overall mean on both `PickCube-v1` and `StackCubeEasy-v1`.

| Metric | Method | PickCube-v1 | PushCube-v1 | StackCubeEasy-v1 |
|---|---|---|---|---|
| Mean (80 policies) | NNiT (Ours) | **78.8**% | **74.7**% | **54.9**% |
| | D2NWG | 15.6% | 60.0% | 12.2% |
| Top-10 | NNiT (Ours) | **98.8**% | **100**% | **86.4**% |
| | D2NWG | 59.4% | 89.2% | 41.8% |
| Worst Architecture | NNiT (Ours) | **64.8**% | **59.6**% | **42.6**% |
| | D2NWG | 5.6% | 41.2% | 2.0% |

## E. Implementation Details

Specific training hyperparameters and optimization settings are detailed in Table 8.

## F. Extended Performance Analysis on Unseen Architectures

Table 2 follows the standard top-10 protocol of (Liang et al., 2024; Sch"urholt et al., 2024; Soro et al., 2024). To verify that the gap over D2NWG is distributional rather than a selection effect, we additionally report the full 80-policy distribution on the unseen architectures (Table 9).

NNiT's worst architecture on `PickCube-v1` (64.8%) exceeds D2NWG's overall mean (15.6%), so top-10 selection accounts for only a small share of the gap. The three tasks also vary in difficulty according to their success tolerance. `PushCube-v1` allows 0.10 m, `PickCube-v1` requires 0.025 m, and `StackCubeEasy-v1` demands 0.02 m. Tighter tolerances leave less margin for weight error and therefore demand more precise weight generation. D2NWG's collapse tracks this difficulty, with its mean success dropping from 63% on seen architectures to 16% on the unseen `PickCube-v1`, and from 58% to 12% on `StackCubeEasy-v1`. Even on `PushCube-v1`, the most forgiving task, D2NWG loses 21 percentage points, indicating the benchmark is not saturated for width generalization.

## G. Single Unseen Architecture Analysis

We investigate the potential for mode collapse by generating 100 MLPs conditioned on a previously unseen topology with hidden dimensions $[16, 64, 64, 32]$. We assess potential mode collapse by calculating the pairwise Euclidean distances and cosine similarities among the top-10 performing policies, ensuring the model preserves diversity even within optimal solutions.

Figure 6 illustrates the pairwise diversity metrics. The heatmaps show high Euclidean distances and low cosine similarities,

confirming that NNiT generates distinct weights for the identical architectures. We note a single instance of redundancy, which we attribute to stochastic sampling variance given the limited scale of the training set.

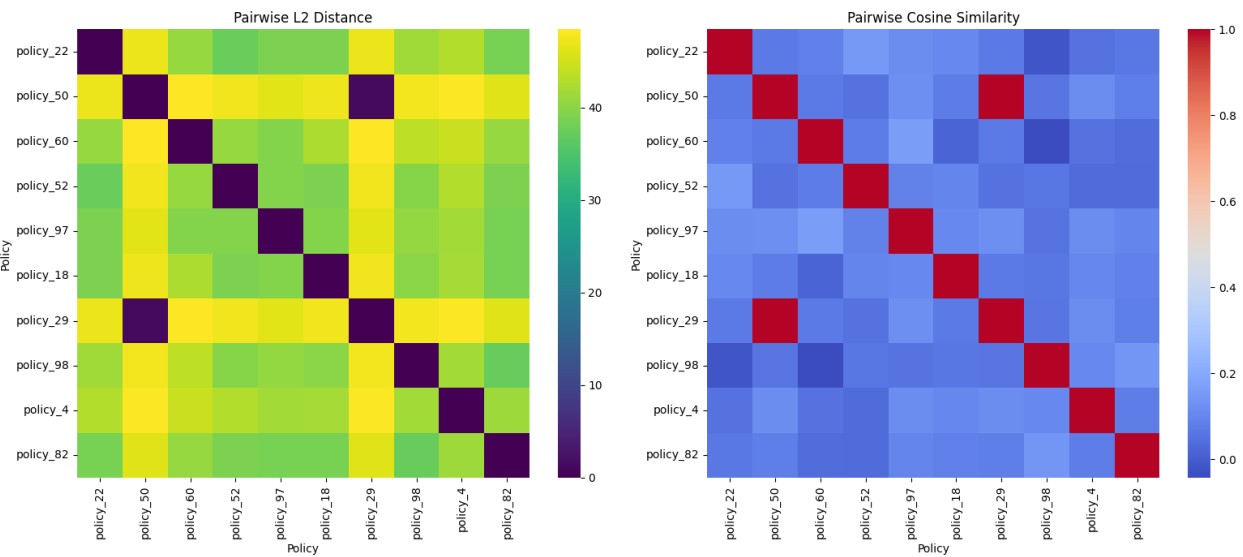

*Figure 6.* **Parametric Diversity Analysis.** Pairwise Euclidean distance (left) and cosine similarity (right) matrices for the top 10 generated policies. The dominance of high distances and low cosine similarities confirms that the generative model preserves variance within the parameter space, avoiding mode collapse.

## H. Validation on the MNIST Benchmark

MNIST model zoos are a standard benchmark for weight-space representation learning, so we additionally evaluate NNiT in this setting. We reuse the same GHN data generation, patch tokenization, width vocabulary $\mathcal{V} = \{\text{input}, 16, 32, 64, \text{output}\}$, and 4-hidden-layer MLP family used in our main experiments, changing only the task.

**Dataset construction.** MNIST's 784-dimensional input far exceeds the input scale our setting supports, so we downsample it to 32 dimensions via PCA fit on the training split. The GHNs are then trained on this family, and the top-100 validated networks per architecture are retained.

**Results.** Across 80 generated networks on the 8 held-out architectures used in our main experiments, NNiT attains 96.1% mean test accuracy, with every network exceeding 80%. The pipeline therefore extends beyond robotic control, since the spatial consistency that NNiT exploits is a property of GHN-generated weight populations rather than of any task-specific bias.

## I. Architecture Diversity under Joint Sampling

For joint sampling, we draw 100 unconditional samples from $p(\mathbf{a}, \mathbf{w})$ on `PickCube-v1`. The model produces 53 unique architectures spanning the $3^4$=81 width configurations, including 3 of the 8 held-out test topologies. Before selection, 97% of generated policies are functional. After top-10 selection, 9 distinct topologies remain with 99% mean success. The diversity observed under fixed-architecture sampling therefore carries over to unconditional joint generation.

## J. Qualitative Policy Rollouts

In this section, we provide qualitative visualizations of the MLPs generated by NNiT. We visualize the execution of the generated MLPs across the three environments. These rollouts demonstrate that the sampled MLPs successfully capture the functional logic required for delicate robotic manipulation.

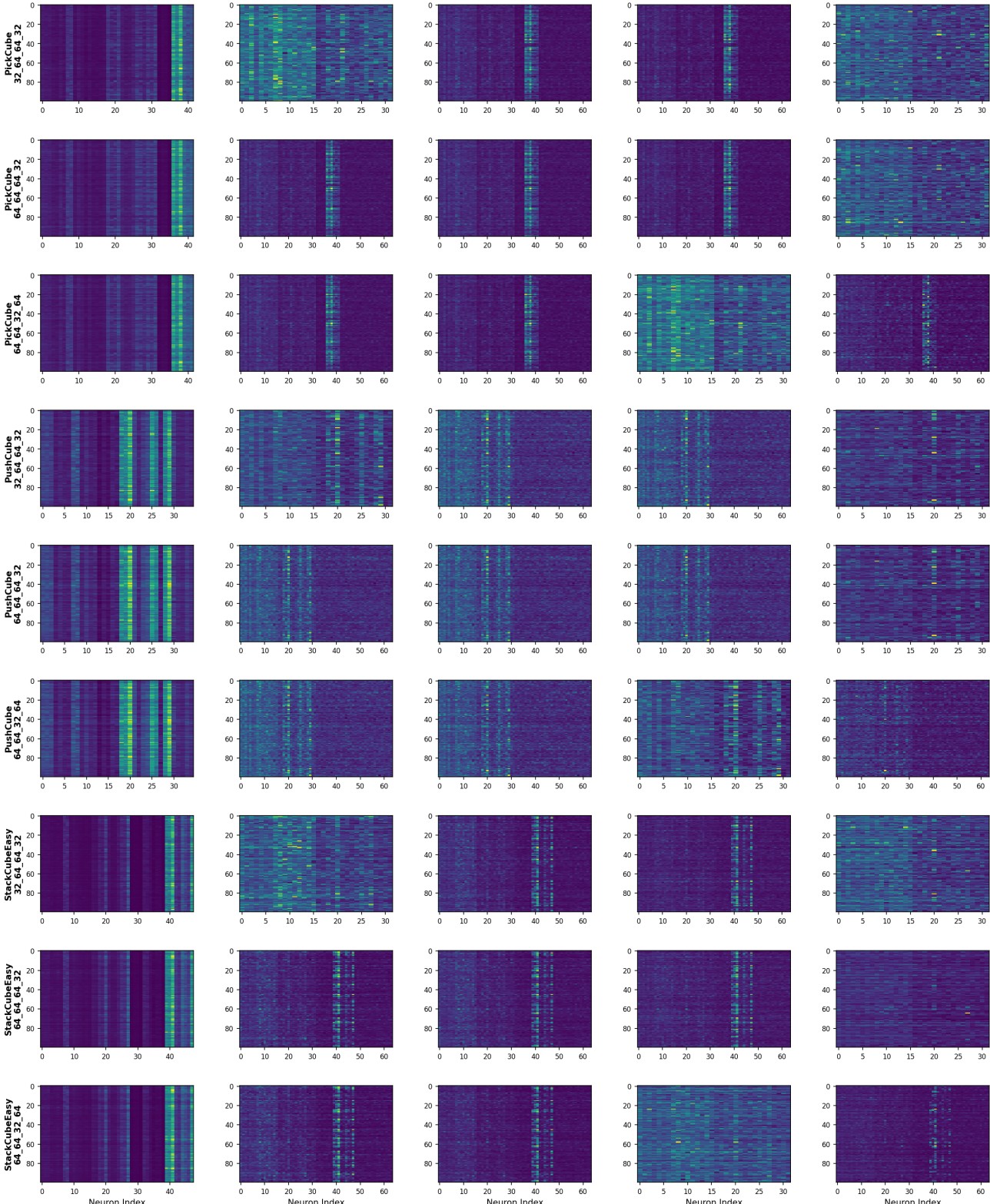

*Figure 5.* **Visualization of Topological Anchoring.** Heatmaps of neuron-wise weight magnitudes for 3 selected architectures across 100 filtered seeds. The vertical banding visually confirms the induction of spatial correlation, validating the premise that these weights can be treated as continuous fields.

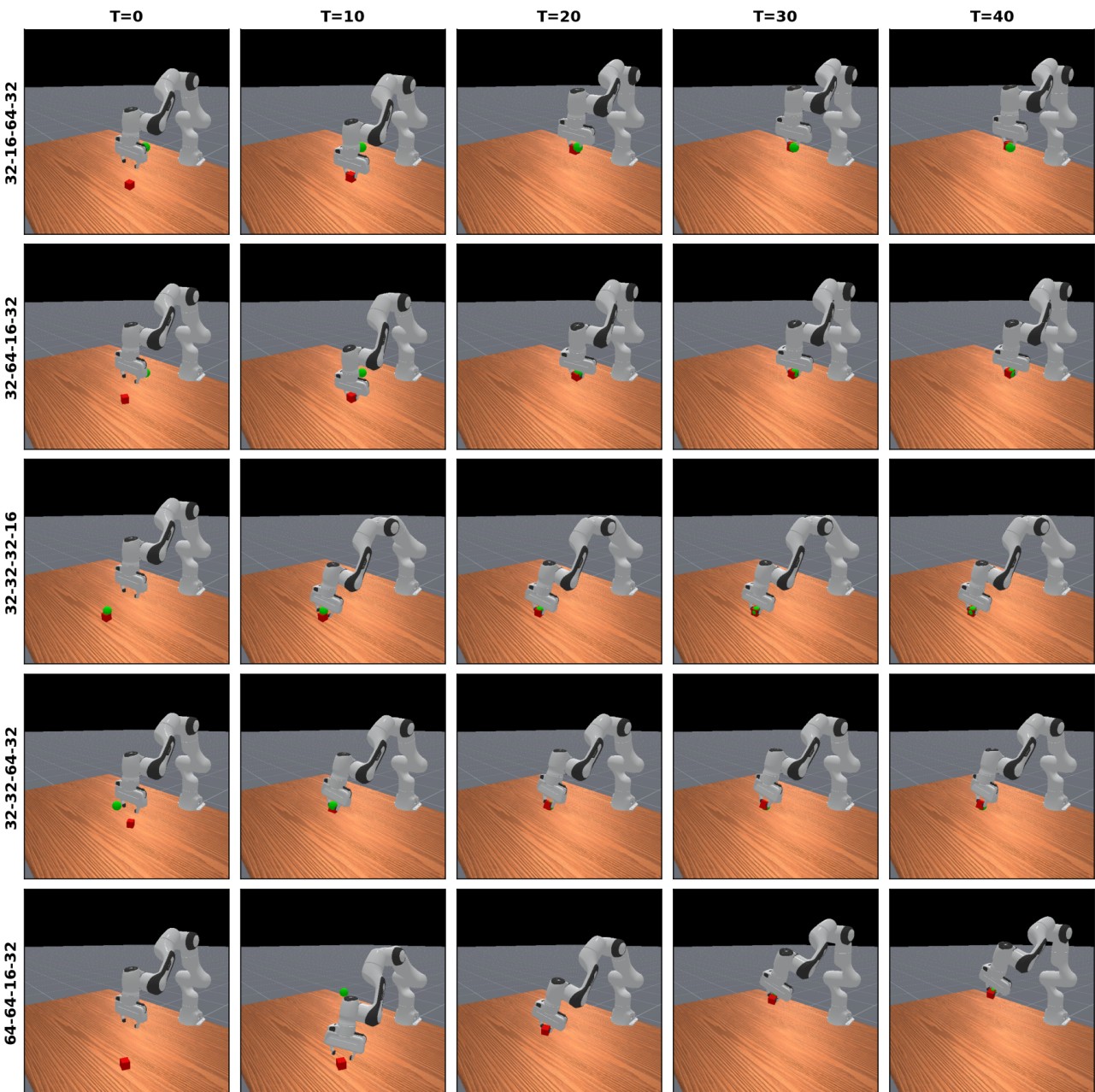

*Figure 7.* **PickCube-v1 Policy Rollout.** A sequential visualization of policies generated by NNiT across diverse topologies. **Left (Y-axis):** Target architecture configurations denoted by their hidden layer widths (e.g., 32-16-64-32). **Top (X-axis):** Temporal snapshots of the rollout in 10-step increments. The agent successfully navigates to the target object, executes a stable grasp, and lifts the cube to the designated goal, visually demonstrating that the sampled NNiT MLPs operate correctly.

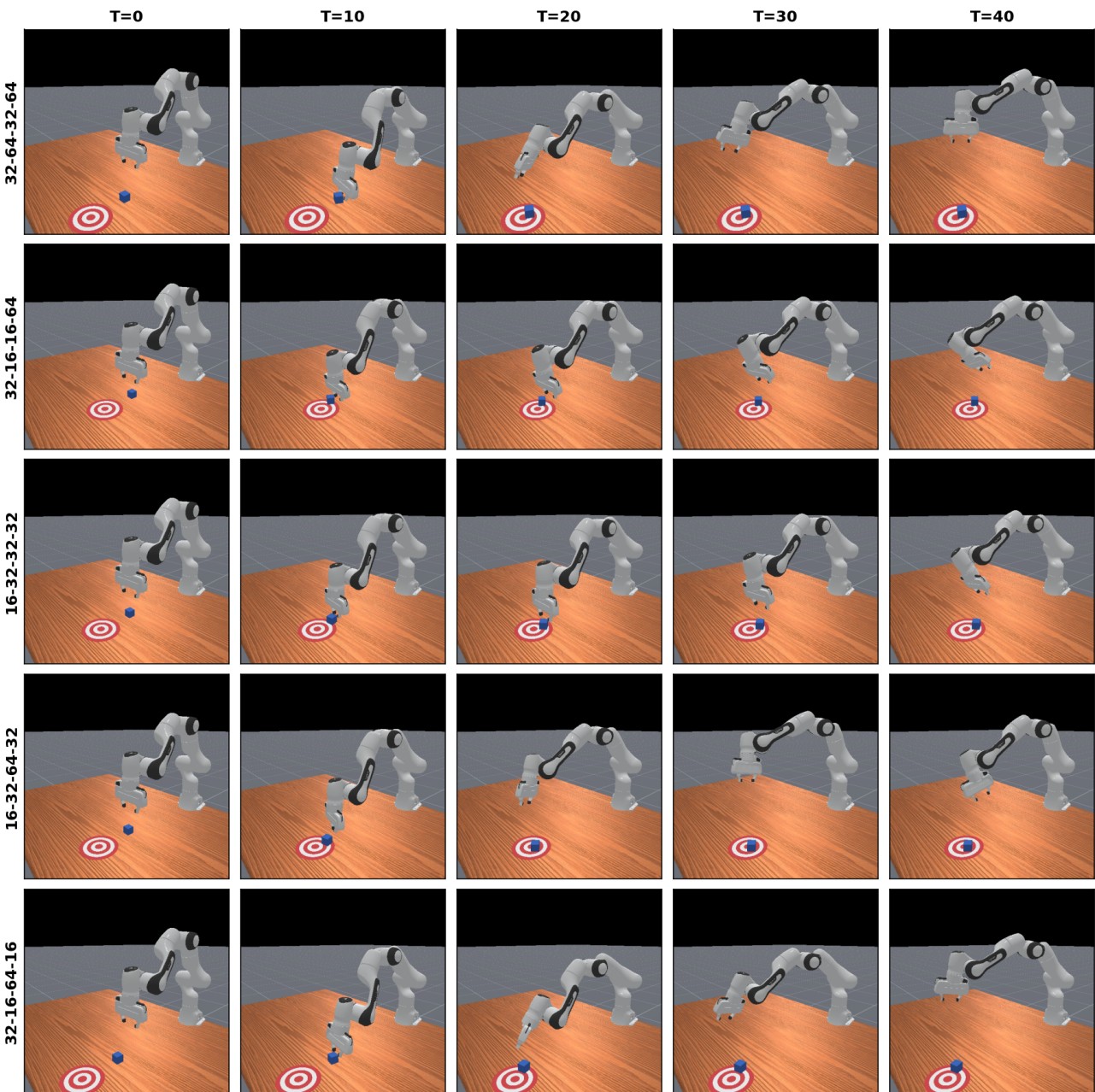

*Figure 8.* **PushCube-v1 Policy Rollout.** A sequential visualization of policies generated by NNiT across diverse topologies. **Left (Y-axis):** Target architecture configurations denoted by their hidden layer widths (e.g., 32-64-32-64). **Top (X-axis):** Temporal snapshots of the rollout in 10-step increments. The policy effectively coordinates the end-effector to manipulate the object, pushing it into the target goal region.

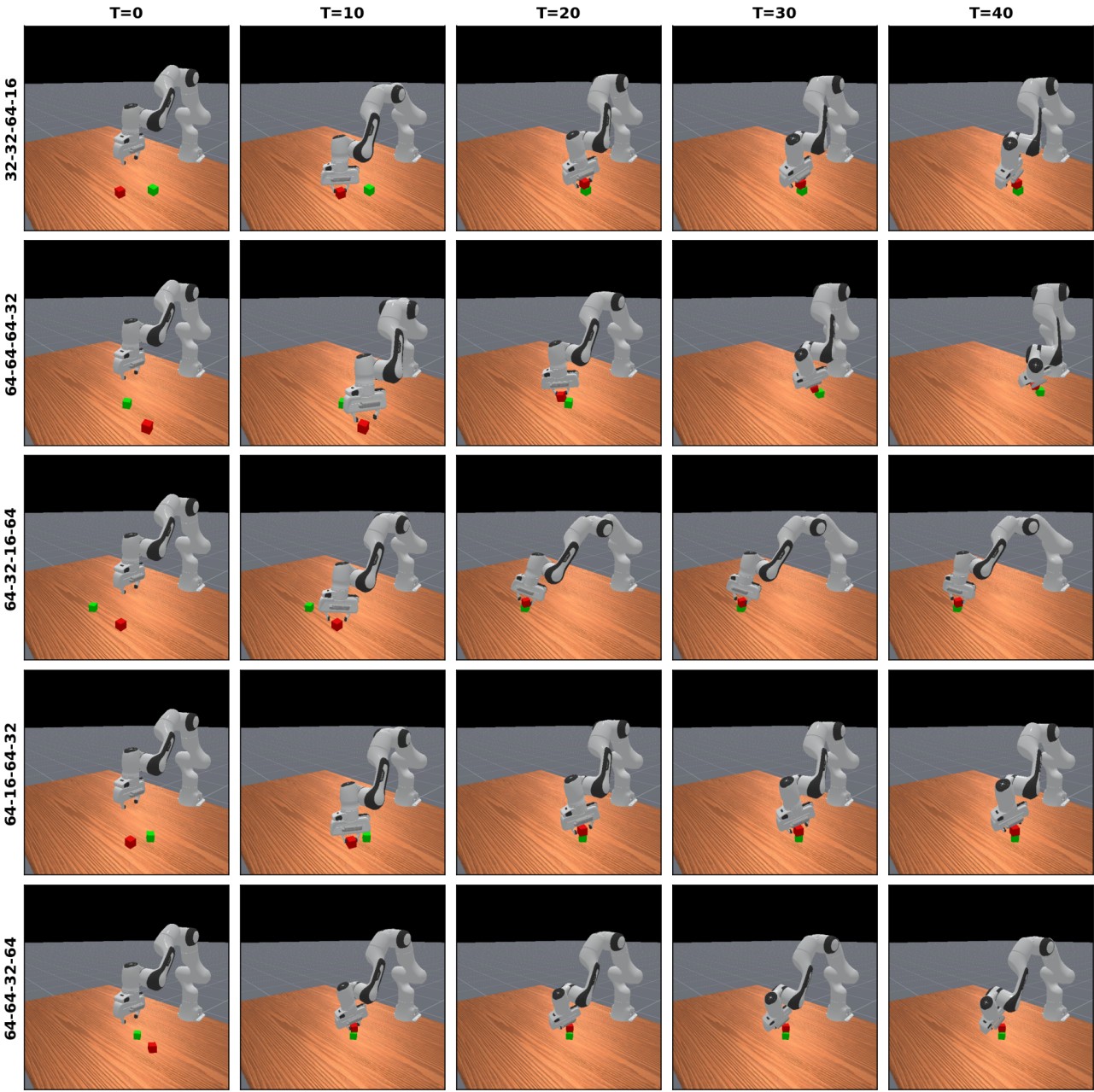

*Figure 9.* **StackCubeEasy-v1 Policy Rollout.** A sequential visualization of policies generated by NNiT across diverse topologies. **Left (Y-axis):** Target architecture configurations denoted by their hidden layer widths (e.g., 64-64-64-32). **Top (X-axis):** Temporal snapshots of the rollout in 10-step increments. The synthesized agent demonstrates high-precision control by successfully grasping the red cube and placing it on top of the green goal cube.

