# OpenReview forum: "NNiT: Width-Agnostic Neural Network Generation with Structurally Aligned Weight Spaces"
_ICML.cc/2026/Conference — ICML 2026 regular_

### Official Review · Reviewer_UKxj · 2026-03-12

**Soundness:** 2
**Presentation:** 3
**Significance:** 2
**Originality:** 3
**Overall Recommendation:** 4
**Confidence:** 4

**Summary:**

This submission proposes a new method called NNiT to generate weights for policies. In particular, the authors identify a gap in related work in generalization across layer width. To achieve better width generalization, they propose to use graph hypernetworks instead of regular SGD training as implicit canonicalization of populations. They encode the architecture in a sequence of discrete token, patchify the weights of entire populations as one and train a Diffusion Transformer on the resulting token sequence. In experimental evaluation, NNiT outperforms existing work on three cube manipulation tasks.

**Compliance With Llm Reviewing Policy:**

Affirmed.

**Final Justification:**

The rebuttal has addressed some of my concerns, although some remain w.r.t to the generality of the method, the evaluation method and experiments. Overall, the submission does add to the communities understanding of how to achieve some aspects of generalization in weight space learning

**Key Questions For Authors:**

One of the main issues with weight space learning is lack of canonical benchmarks. While I appreciate the addition of a new task, it makes contextualization with existing work challenging. Is there a reason why the authors chose this particular task, and have they considered comparing their method on existing WSL tasks?

**Limitations:**

Yes

**Strengths And Weaknesses:**

## Strengths

- Using GHNs as a way to condition models akin to canonicalization is an interesting idea.
- Generalization is a highly relevant topic in weight space learning, and width generalization a necessary step towards broader generalization.
- The architecture and weight encoding is straight-forward and seems functional for width generalization.
- The results show good performance on the tested tasks, and indicate that NNiT does indeed generalize across widths.

## Weaknesses
- The authors use GHN as a training method for implicit canonicalization, which I find an intriguing idea. One caveat is that it requires models to be trained from scratch, it does not apply to existing populations. Further, looking at Figure 3 and Table 1, I’m not sure that I follow the authors conclusion that the results are canonicalization and not just mode collapse. Weight similarity is hard to estimate. There are two other angles that might add more information to it: comparing behaviorally (CKA, SVCCA, agreement, etc.) between the models. If GHN indeed does not collapse, then the behavioral similarity of GHN generated models should be ± the same as the similarity of trained models. Further, the authors could apply git re-basin like canonicalization methods on both SGD and GHN models, as well as apply random permutations (aligning or dis-aligning) to estimate if that shows a similar effect as GHN vs SGD trained models. Also, an ablation training NNiT on SGD vs GHN populations to evaluate the effect on the synthesized models would be interesting.
- The authors identify permutation symmetry as a central challenge, which I intuitively agree with. However, recent work indicates that pre-training weight representation methods at scale makes dealing with permutations if not irrelevant, at least less pressing, (e.g. [2]). Scaling also appears to naturally lead to generalization.
- The authors focus on generalization across architectures and widths, but it seems the patchification is tied to a single architecture. If widths between layers vary a lot, padding will dominate for some of the patches. Are padding-only patches removed, or is this considered during training? Previous work has explicitly investigated the impact of uneven signal density of weight tokens and found that it can be a source of inefficiency and instability for training [1, 3].
- The authors propose a method that can generalize across architectures and widths (lines 78,79). The evaluation seems limited to MLPs of the same architecture, varying only the width of the hidden layers. While that is absolutely fine to test a method, it is a more narrow than what the introduction suggests.

[1] Wang et al. Scaling Up Parameter Generation: A Recurrent Diffusion Approach, Neurips 2025.
[2] Falk et al., Learning Model Representations Using Publicly Available Model Hubs, 2025.
[3] Falk et al., The Impact of Model Zoo Size and Composition on Weight Space Learning, 2025.

---

> ### Author Rebuttal · Authors · 2026-03-31
>
> We appreciate the recognition that GHN-based conditioning is an interesting idea (Strength (S)-1), that width generalization is a necessary step toward broader generalization (S-2), and that NNiT does indeed generalize across widths (S-4).
>
> We also thank the reviewer for the detailed evaluation and for the specific constructive suggestions (CKA comparison and permutation ablation). We take the suggestions in stride and implement them below:
>
> **[Weakness (W)-1]**
>
> >``I'm not sure... the results are canonicalization and not just mode collapse... comparing behaviorally (CKA, SVCCA)... apply random permutations.''
>
> Following the reviewer's suggestions directly, we added two new ablations:
>
> **(1) CKA representational comparison (new).**
>
> Pairwise linear CKA across 35 GHN and 35 SGD MLPs (PickCube-v1):
>
> | | **GHN (35)** | **SGD (35)** |
> |---|---|---|
> | Overall CKA | 0.849 ± 0.088 | 0.933 ± 0.030 |
> | Layer 1 | 0.917 ± 0.016 | 0.957 ± 0.005 |
> | Layer 2 | 0.880 ± 0.033 | 0.926 ± 0.022 |
> | Layer 3 | 0.751 ± 0.083 | 0.918 ± 0.037 |
>
> GHN exhibits *lower* CKA at every layer, indicating greater representational diversity than SGD. If GHN were mode-collapsing, we would expect CKA to be higher than SGD, not lower. The effect is strongest in deeper layers (Layer 3: 0.751 vs. 0.918), where GHN models explore the diverse representations.
>
> These results confirm that GHN produces spatial alignment without mode collapse.
>
> **(2) Spatial alignment ablation (new).**
>
> We apply random neuron permutations to GHN-generated weights, which preserve weight statistics but destroy the 2D spatial ordering. Training NNiT on these permuted weights collapses performance from 78.8\% to 0\% (0/80 policies on PickCube-v1). This mirrors SANE's failure mode: when positional encoding no longer corresponds to consistent weight structure, the generative model cannot learn.
>
> We can extend this further. Cross-seed magnitude correlation is $r \approx 0.65$ for intact GHN weights, but drops to $r \approx 0.00$ for both permuted GHN and SGD-trained weights. This permutation ablation essentially serves as a controlled approximation of SGD generation, while retaining weight statistics and letting us isolate the effect of spatial alignment directly.
>
> Post hoc methods like Git Re-Basin align networks within a single architecture but yield topology-specific alignment. GHN provides cross-architecture spatial consistency from a single pretrained model, which is a novel result.
>
> Taken together, CKA confirms that GHN weights are representationally diverse. The permutation ablation confirms that GHN's spatial alignment is required. In doing so, GHN provides a consistent spatial structure that the DiT can learn from, without collapsing the weight population to a narrow set of solutions.
>
> **[W-2]**
>
> >``Recent work indicates that pre-training weight representation methods at scale makes dealing with permutations... at least less pressing [2].''
>
> Scaling and structural alignment represent complementary directions. [2] builds on the SANE with the same $P{=}[n,l,k]$ positional encoding and 1D tokenization that fails on our task (0--6\%, Table~2). **See Reviewer PXCs Q-3 response for why SANE fails.** These works also don't show unseen width combinations or zero-shot deployment, requiring several epochs of fine-tuning.
>
> **[W-3]**
>
> >``If widths between layers vary a lot, padding will dominate... [1, 3] found uneven signal density can be a source of inefficiency.''
>
> Prior work [1,3] only focused on masking. In NNiT, architecture tokens $[w_1,\ldots,w_L]$ are modeled jointly with weight patches. Since we are using 2D patches, padded regions are clustered much better than if we flatten them into a 1D sequence. So, the architecture conditioning defines a deterministic crop at deployment and can infer padding well (Section~5.4). Empirically, training remains stable even for width-16 layers with substantial padding.
>
> **[W-4, Question (Q)]**
>
> >``The evaluation seems limited to MLPs... more narrow than what the introduction suggests... Is there a reason why the authors chose this particular task?''
>
> Thank you so much for pointing this out. We agree and will tighten this framing in the camera-ready.
>
> Our contribution is width generalization within a fixed depth and fixed vocabulary. There are already WSL benchmarks for depth transfer where prior methods perform well. So, we focus on the complementary open problem of width-diverse generation, for which no standard benchmark exists. We chose robotic manipulation because policy quality is functionally verified: weight errors compound through the control loop, especially under tight tolerances (0.025/0.02\,m).
>
> In response to other reviewers, we additionally tested MNIST classification, achieving 96.1\% mean accuracy across unseen architectures, confirming the representation transfers beyond ManiSkill3.

---

> > ### Author Rebuttal · Reviewer_UKxj · 2026-04-04
> >
> > I would like to thank the authors for their response. I agree with the authors, adding the CKA comparison supports their case for using GHNs as data generators. Adding the permutation test confirms that picture. I'm not sure I agree with W2, existing results explicitly include generalization to different widths as a subset of general variation. This choice does make sense if the base models have to be trained. As some of the other reviewers, I wonder if the evaluation design complicates the comparison to baselines. Existing methods focus on the representation learning aspect of generalization, while the authors appear to focus on generated performance in particular. I encourage the authors to revise that section towards understanding the methods, less beating a baseline. W3, glad to hear it doesn't affect stability, although I maintain it affects information density. W4, generalization to MNIST is a promising signal. I encourage the authors to continue on that path to strengthen the experimental setup and empirical evidence. With that in mind, I increase my score to weak accept.

---

> > > ### Author Response · Authors · 2026-04-06
> > >
> > > Thank you for the thoughtful comments. We’re glad to see that we addressed most of your questions. We also appreciate your point about revising our explanations and comparing representations instead of only focusing on generated performance differences. We will take that advice in stride as we edit and improve our manuscript. Thanks again for the review and helping us improve our paper. Authors

---

### Official Review · Reviewer_6KGn · 2026-03-12

**Soundness:** 2
**Presentation:** 3
**Significance:** 2
**Originality:** 2
**Overall Recommendation:** 4
**Confidence:** 2

**Summary:**

The paper proposes Neural Network Diffusion Transformers (NNiTs), a generative framework for neural network parameter synthesis. To address the challenges of permutation symmetry and fixed-dimensional weight representations, the authors utilize Graph HyperNetworks (GHNs) with a CNN decoder to induce structural alignment in the weight space. By treating weights as a continuous spatial field and employing patch-based tokenization, NNiT enables the zero-shot synthesis of functional weights for architectural widths and topologies unseen during training.

**Compliance With Llm Reviewing Policy:**

Affirmed.

**Final Justification:**

The rebuttal addressed my concern.

**Key Questions For Authors:**

- Given that the generative network is trained on a dataset with predefined weight shapes, how does the performance deteriorate when it is used to generate networks with weight shapes vastly different from the training set (For example, weight with hidden widths of 128, 256, 1024).

- Can the authors provide training and inference time metrics of the experiment to demonstrate the efficiency of the approach?

- Can the author consider evaluating NNiT on more complex datasets or tasks that do not have saturated performance to further validate its robustness?

**Limitations:**

yes

**Strengths And Weaknesses:**

**Strengths**
- The patch-based tokenization effectively decouples weight generation from fixed matrix dimensions, enabling zero-shot synthesis for unseen architectural widths.

- Utilizing Graph HyperNetworks (GHNs) with CNN decoders successfully induces structural alignment and local correlation in the weight space.

- The framework unifiedly models discrete architecture tokens and continuous weight patches, allowing for both joint co-design and conditional synthesis.


**Weaknesses**

- The core methodology is primarily an integration of existing primitives such as Diffusion Transformers (DiT), GHNs, and the MoNL framework.

- The empirical evaluation is limited to robotic manipulation tasks within the ManiSkill3 environment. To further demonstrate the robustness and generalizability of the width-agnostic approach, can you consider conducting an experiment on other domains like image recognition (MNIST or CIFAR-10).

- Success rates reaching near 100% on ManiSkill3 tasks suggest the current benchmarks may be too simple and saturated to show the benefit of the proposed method compared to other baselines.

---

> ### Author Rebuttal · Authors · 2026-03-31
>
> We thank the reviewer for recognizing the effectiveness of patch-based tokenization for decoupling generation from fixed dimensions (Strength (S)-1), GHN-induced structural alignment (S-2), and the unified joint modeling of architecture and weights (S-3). We address each concern below.
>
> **[Weakness (W)-1]**
>
> >``The core methodology is primarily an integration of existing primitives.''
>
> Our main contribution is not in any single component but in identifying a previously unrecognized structural property of GHN weights - *consistent 2D spatial structure across seeds* (cross-seed correlation $r \approx 0.65$ vs. $r \approx 0.00$ for SGD), and building a framework that exploits it. This property enables *2D patch tokenization where width variation becomes resolution variation*, a perspective that does not exist in prior work. We verify this through controlled ablation: destroying the spatial structure via random neuron permutations or using SGD-style weights collapses NNiT from 78.8\% to 0\%.
>
> We can also observe that D2NWG, using the same GHN data, still degrades on unseen architectures. Without GHN alignment, NNiT collapses to 0\%. Without patch tokenization, D2NWG drops to 15.6\%. It is this combination that constitutes our contribution, width generalization that no prior method achieves.
>
> **[W-2, Question (Q)-3]**
>
> >``The empirical evaluation is limited to robotic manipulation tasks... can you consider conducting an experiment on other domains like image recognition (MNIST or CIFAR-10)?"
>
> Following this suggestion directly, we evaluated NNiT on **MNIST digit classification** using an identical pipeline (GHN, patch tokenization, width vocabulary {input,16,32,64,output}, 4-hidden-layer MLPs) with no modifications. Because raw MNIST inputs are 784-dimensional, we apply PCA to 32 dimensions to match the scale of our MLP setting and our time constraints for rebuttal.
>
> | **MNIST (8 unseen architectures)** | **NNiT** |
> |---|---|
> | Mean test accuracy (80 policies) | **96.1%** |
> | Minimum policy accuracy | >80% |
>
> Here, the clear success on MNIST confirms that NNiT's pipeline is domain-agnostic and not limited to robotic tasks. The same architecture, tokenization, and training procedure transfer directly to classification without issue.
>
> We chose MLPs because they naturally require diverse per-layer widths, whereas CNNs (as with the ResNet family) use standardized per-stage channel counts, making depth the primary axis of variation. D2NWG already performs reasonably for depth transfer; NNiT addresses the complementary open problem of width transfer.
>
> **[W-3]**
>
> >``Success rates reaching near 100\% on ManiSkill3 tasks suggest the current benchmarks may be too simple.''
>
> The near-100\% numbers reflect top-10 selection on *seen* architectures and do not capture the full evaluation landscape. On **unseen** architectures (the core evaluation), looking at the full 80-policy test set reveals a clear separation:
>
> | Metric | Method | PickCube | PushCube | StackCubeEasy |
> |---|---|---|---|---|
> | Mean success (unseen) | NNiT | 78.8% | 74.7% | 54.9% |
> |  | D2NWG | 15.6% | 60.0% | 12.2% |
> | D2NWG seen→unseen drop | | −47.6pp | −21.0pp | −46.0pp |
>
> The three tasks also form a strict difficulty gradient (PushCube: 0.1m tolerance, PickCube: 0.025m, StackCubeEasy: 0.02m). D2NWG collapses from 63\%$\to$16\% (PickCube) and 58\%$\to$12\% (StackCubeEasy) on unseen architectures. Despite PushCube having the most forgiving tolerance, D2NWG's 21pp drop still demonstrates the task is far from saturated for width generalization. While SANE fails entirely (0--6\%).
>
> **[Q-1]**
>
> >``How does the performance deteriorate... with hidden widths of 128, 256, 1024?''
>
> NNiT currently generates weights for unseen *topologies* like novel combinations of known widths from {16, 32, 64}, not for unseen width values. Currently, extending to new unseen widths (e.g., 128) would require GHN training data at that width and a corresponding discrete token. A wider layer simply produces more patches. As with FiT for higher-resolution images, widening the vocabulary is primarily a data constraint, not a model limitation. We foresee that our method can extrapolate to these larger unseen widths, but that is not the focus of our current NNiT paper and would require significantly more compute and time overhead to properly evaluate.
>
>
> **[Q-2]**
>
> >``Can the authors provide training and inference time metrics?''
>
> Training: **30 hours** on 4$\times$A6000 GPUs.
> Sampling: $\sim$ **40s** per policy on a single GPU (1000 DDPM steps).
> Generation is a one-time offline cost; the deployed MLP runs at standard inference speed.

---

> > ### Author Rebuttal · Reviewer_6KGn · 2026-04-03
> >
> > Thank you for the clarifications. The rebuttal has addressed my concerns, and I have accordingly updated my score to 4.

---

> > > ### Author Response · Authors · 2026-04-03
> > >
> > > Thank you for your comments. We're glad to see that our clarification and additional study have addressed your questions. We will revise our paper accordingly and incorporate the added studies. Thank you again for your review.
> > >
> > > Authors

---

### Official Review · Reviewer_PXCs · 2026-03-12

**Soundness:** 3
**Presentation:** 3
**Significance:** 2
**Originality:** 3
**Overall Recommendation:** 4
**Confidence:** 4

**Summary:**

This paper is about generative modeling of neural network weights under changing layer widths. The core idea is to first use Graph HyperNetworks (GHNs) with a CNN decoder to induce a more structurally aligned weight space, and then to model these aligned weights with a diffusion transformer that tokenizes the weights into local patches instead of flattening them into a fixed-size vector.

The resulting method, NNiT, jointly models discrete architecture width tokens and continuous weight patches, enabling both architecture-conditioned weight synthesis and joint architecture-weight generation. Empirically, the paper focuses on MLP policies for ManiSkill3 robotic control, constructs a dataset of GHN-generated expert policies over 64 training and 8 held-out architectures, and reports that NNiT maintains strong zero-shot performance on unseen widths/topologies while D2NWG degrades and SANE fails in this setting. The paper also presents qualitative evidence that GHNs induce locally correlated, more consistent weight layouts than SGD-trained models, which is used to justify the patch-based representation.

**Compliance With Llm Reviewing Policy:**

Affirmed.

**Final Justification:**

Having read the author's rebuttal, I decided to raise my rating.

**Key Questions For Authors:**

**Q-1**: The reported results use the top-10 policies selected from 100 generated samples. Can the authors also report top-1 performance and summarize the full performance distribution over the 100 samples for NNiT and D2NWG? A strong answer here would help me understand whether the method is robust at the sample level or whether the current gains depend heavily on selection.

**Q-2**: Since the central premise is that GHN + CNN decoding induces the local structure needed for patch-based generation, can the authors clarify how essential this ingredient is in practice? For example, do preliminary experiments with SGD-trained weights, or with another alignment/canonicalization method, fail completely or only degrade gracefully?

**Q-3**: SANE appears to fail even on seen architectures in this setup, while D2NWG remains competitive on seen architectures but drops on unseen ones. Can the authors provide more detail on the exact baseline adaptations, including tokenization, positional encoding, and hyperparameter choices, and explain why SANE is so weak here beyond what is written in the paper?

**Q-4**: The unconditional joint-generation results are interesting, but the search space is still very small. Can the authors quantify the diversity of generated architectures more explicitly, e.g., frequency of unseen topologies under joint sampling, or the fraction of sampled architectures that are functional before top-k selection? A stronger analysis here would help support the claim that the model has learned structural logic rather than mainly memorizing a small, discrete family.

**Q-5**: The current results are limited to up to four hidden-layer MLPs, which the limitations section acknowledges. Without asking for a major new experiment, it would be helpful if the authors could clarify what they believe is the first true failure mode when scaling: sequence length, weight dynamic range, data scarcity, or something else. This would make the limitations section more concrete and help readers judge how close the approach is to broader applicability.

**Limitations:**

yes

**Strengths And Weaknesses:**

**S-1**: I think this is an interesting paper with a creative core idea. The combination of GHN-induced alignment and patch-based diffusion over weights is novel enough to stand out from prior weight-generation work that mostly relies on flattened vectors or latent embeddings. I also like the central intuition: if weight spaces can be made locally structured, then width changes can be treated more like resolution changes, which is a clean and potentially useful perspective.

**S-2**: The paper is easy to follow with helpful figures able to convey the method's core functionality. The empirical claims are meaningful as the experimental evaluation uses functional robotic control rather than only proxy metrics on static tasks. The zero-shot architecture-conditioned results are also genuinely promising: on network topologies with unseen widths, NNiT remains close to dataset-level performance, whereas D2NWG drops noticeably on two tasks and SANE fails throughout this setup.

**S-3**: Another strength is that the paper tries to motivate the representation choice empirically rather than simply assuming it. The visualization on page 6 and the diversity analyses in the appendix do support the claim that GHN-generated policies exhibit repeatable spatial patterns without obvious mode collapse. I also appreciate that the paper does not only study conditional synthesis (p(w|a)), but additionally evaluates joint synthesis (p(a,w)), and reports that the unconditional model can generate functional policies, including on some held-out architectures.

**Conclusion 1**: I do think there is a real contribution here in reframing "network width" generalization as a tokenization problem over aligned parameter fields.

**W-1**: My main concern is that the current empirical scope is still quite narrow relative to the paper’s framing. The entire story is built on small MLP policies in ManiSkill3, with widths drawn from a tiny discrete vocabulary {16, 32, 64}, 64 training architectures, and only 8 held-out test topologies. More importantly, the training corpus is not a broad population of independently trained models, but a filtered corpus of GHN-generated policies, where the top-100 validated policies per architecture are retained, yielding 6,400 expert policies in total. This makes the paper feel less like a general solution to width-agnostic neural network generation and more like a proof of concept for generative modeling on a specific GHN-induced policy family. I think this distinction matters a lot.

**W-2**: A related weakness is that it is not yet fully clear how much of the improvement comes from NNiT itself versus from the special structure of the GHN-generated data. The paper shows qualitative alignment differences between GHN and SGD weights, but it does not really isolate the necessity of each ingredient: GHN alignment, patch tokenization, multimodal joint training, and the particular MoNL setup. Since all baselines are trained on the same GHN-aligned corpus, and since SANE fails even on seen architectures, I found it hard to judge how much of the empirical narrative reflects a genuinely stronger method versus how much reflects baseline mismatch or sensitivity to the chosen representation. The paper states that the baselines were adapted to this setting because prior work does not target zero-shot architectural generalization, which is fair, but it also means the comparison is less decisive than the headline might suggest.

**W-3**: I also have some reservations about the evaluation protocol. The main reported metric is the performance of the top-10 policies selected from 100 generated samples, motivated as an offline validation phase before deployment. I understand the motivation, but this is still a fair selection protocol, especially in a setting where architecture-conditioned generation is supposed to be practically useful. I would have liked to see more visibility into the full sample distribution, top-1 performance, and sample efficiency. Similarly, the joint generation results are interesting, but there are no baselines there, and the architecture-diversity evidence remains rather limited given the small search space.

**Conclusion 2**: Overall, I think the paper is technically reasonable and contains a good idea, but the current evidence is not yet strong enough for me to be fully convinced by the broader claims.

---

> ### Author Rebuttal · Authors · 2026-03-31
>
> Thank you for the thorough evaluation and for recognizing the creative core idea (Strength (S)-1), the meaningful empirical evaluation using functional robotic control (S-2), and that there is a real contribution in reframing width generalization as a tokenization problem over aligned parameter fields (Conclusion~1). We address each concern below.
>
> **[Weakness (W)-3, Question (Q)-1]**
>
> >``I would have liked to see more visibility into the full sample distribution, top-1 performance, and sample efficiency."
>
> We agree this is important and would like to provide these details now. Our framework's strength is best demonstrated by looking at the full 80-policy test set (10 policies $\times$ 8 unseen architectures) rather than isolating top-K:
>
> | Metric | Method | PickCube | PushCube | StackCubeEasy |
> |---|---|---|---|---|
> | Mean (80 pol.) | NNiT | 78.8% | 74.7% | 54.9% |
> |  | D2NWG | 15.6% | 60.0% | 12.2% |
> | Top-10 | NNiT | 98.8% | 100% | 86.4% |
> |  | D2NWG | 59.4% | 89.2% | 41.8% |
> | Worst-arch | NNiT | 64.8% | 59.6% | 42.6% |
> |  | D2NWG | 5.6% | 41.2% | 2.0% |
>
>
> The gap is distributional, not a selection artifact. NNiT's worst single architecture still outperforms D2NWG's mean on PickCube and StackCubeEasy; while SANE fails entirely (0--6\%).
>
>
> **[W-2, Q-2]**
>
> >``It is not yet fully clear how much of the improvement comes from NNiT itself versus from the special structure of the GHN-generated data.''
>
>
> We isolate both components with controlled ablations:
>
> **(1) GHN alignment is necessary.**
>
> We apply random neuron permutations to GHN-generated weights, which preserve weight statistics but destroy the 2D spatial ordering. Training NNiT on these permuted weights collapses performance from 78.8\% to 0\% (0/80 policies on PickCube-v1), mirroring SANE’s failure when positional encoding no longer aligns with weight structure under width variation. This ablation also closely matches SGD: cross-seed magnitude correlation is $r \approx 0.65$ for intact GHN weights, but drops to $r \approx 0.00$ for both permuted GHN and SGD-trained weights. This makes the ablation a controlled approximation of SGD-style weights, while retaining weight statistics and letting us isolate the effect of spatial alignment directly.
>
> To also address Q-2, post hoc methods like Git Re-Basin align networks within a single architecture but yield topology-specific alignment. GHN provides cross-architecture spatial consistency from a single pretrained model, which is a novel result.
>
> **(2) Patch tokenization is necessary beyond GHN.**
>
> D2NWG is trained on the same GHN data yet degrades severely on unseen architectures (Table~2). Its CLIP-based class-index conditioning requires having seen topologies during training. In contrast, NNiT's width tokens (e.g., $[16,32,64,16]$) allow known widths to be recombined in unseen arrangements.
>
> Neither component alone is sufficient. Without GHN alignment, NNiT collapses to 0\%. Without patch tokenization, D2NWG only achieves 15.6\% on the PickCube test set. It is the combination of both that enables width generalization.
>
> **[Q-3]**
>
> >``Can the authors... explain why SANE is so weak here?"
>
> SANE's positional embedding $P = [n, l, k]$ encodes the global position, layer index, and within-layer row index. Under width variation, this breaks: for architecture $[16,32,...]$, layer~2 starts at $n{=}17$; for $[64,32,...]$, it starts at $n{=}65$. The same $n{=}17$ refers to layer 2 row 1 in one architecture but layer 1 row 17 in another. Combined with SANE's per-layer normalization (which also assumes fixed architectures), this positional misalignment is a fundamental problem and explains why SANE performs poorly in our width-diverse setting.
>
> **[Q-4]**
>
> >``Can the authors quantify the diversity of generated architectures...?''
>
> Under joint sampling ($p(a,w)$, 100 samples, PickCube): **53 unique architectures** generated (of 81 possible), including **3 held-out topologies**. Before selection, **97\%** produce functional policies. After top-10 selection, **9 distinct architectures** remain with 99\% mean success.
>
>
> **[W-1, Q-5]**
>
> >``The current empirical scope is still quite narrow... what is the first true failure mode when scaling?"
>
> We additionally evaluated NNiT on MNIST digit classification using an identical pipeline, achieving **96.1\%** mean accuracy across 80 policies from 8 unseen architectures, with every policy exceeding 80\%. This confirms the pipeline transfers across task families.
>
> We set our scope explicitly on width-agnostic MLP generation, a setting where all prior methods fail. We view this as analogous to early DiT work validating on $64{\times}64$ images. The framework requires no architectural changes for deeper networks (longer sequences) or wider layers (more patches). The main bottleneck at scale is *data generation* with time and compute bottlenecks, not model design.

---

> > ### Author Rebuttal · Reviewer_PXCs · 2026-04-04
> >
> > I would like to thank the authors for their rebuttal. My questions have been answered and I would like to raise my rating to weak accept.

---

> > > ### Author Response · Authors · 2026-04-06
> > >
> > > Thank you for your comments. We're glad to see that our clarification and additional study have addressed your questions. We will revise our paper accordingly and appreciate your advice in helping us improve the manuscript.
> > > Thank you again for your review.
> > > Authors

---

### Official Review · Reviewer_VtiK · 2026-03-18

**Soundness:** 2
**Presentation:** 3
**Significance:** 3
**Originality:** 2
**Overall Recommendation:** 4
**Confidence:** 2

**Summary:**

This paper focuses on weight generation in neural networks, aiming to improve its generalization ability across network architectures with varying widths. The authors primarily address the challenge that existing methods struggle to generalize to unknown widths due to permutation symmetry and fixed-size parameter representations. This paper aims to achieve width-independent weight generation and demonstrates encouraging results, particularly on unknown network architectures.

**Compliance With Llm Reviewing Policy:**

Affirmed.

**Final Justification:**

My concerns are solved to some extent.

**Key Questions For Authors:**

The experiments primarily focused on the relatively simple MLP architecture. Could the proposed method be well extended to more complex architectures, such as CNN or Transformers?

Can the method be applied to standard SGD-trained networks?

**Limitations:**

yes

**Strengths And Weaknesses:**

Strengths:

1.	Clear Motivation. This paper points out a significant limitation of previous work: the weight generation model struggles to generalize to unseen widths. This is a very meaningful problem.

2.	Well-Designed Framework. The combination of block-based weight representation, multimodal diffusion modeling, and GHN-based alignment is very natural. Each component serves a clear objective, and together they address the core problem of width generalization.

3.	Reasonable Experimental Results. This paper achieves significant improvements on unseen architectures compared to previous methods.

Weaknesses:

1．The innovation is lacking. While the overall framework is well-constructed, the components used in this paper (Diffusion Model, Block Tokenization, GHN) are all borrowed from existing work. The main contribution of this paper lies in how to combine these components, rather than introducing entirely new techniques.

2．The experiments primarily focus on relatively simple MLP strategies (ManiSkill3 task). Whether this method can be extended to more complex architectures (CNN or Transformer) remains unclear.

3．The paper points out that the model can learn structural design principles and generalize to unseen architectures through zero-shot testing, but lacks corresponding experimental results to verify this, requiring further analysis or ablation experiments.

---

> ### Author Rebuttal · Authors · 2026-03-31
>
> Thank you for recognizing the clear motivation behind width generalization (Strength (S)-1), the natural design of our framework (S-2), and the significant improvements on unseen architectures (S-3). We address each concern below.
>
> **[Weakness (W)-1]**
>
> >``Innovation is lacking... the main contribution lies in how to combine these components.''
>
> Beyond combining pre-existing components, our main contribution is in identifying a previously unrecognized structural property of GHN weights and building a framework that exploits it. While prior work treats GHNs purely as a data source, we demonstrate that a GHN with a CNN decoder produces weight matrices with consistent 2D spatial structure across seeds (cross-seed magnitude correlation $r \approx 0.65$ vs. $r \approx 0.00$ for SGD). This property yields three practical consequences that constitute genuinely new contributions to the field:
>
> **(1) Removes the need for costly post-hoc alignment (e.g.\ Git Re-Basin).**
>
> **(2) Simplifies data collection since models trained through GHNs are structurally aligned by construction.**
>
> **(3) Enables 2D patch tokenization of MLP weights, where each layer's weight matrix is treated as a spatial field and width variation becomes resolution variation, analogous to FiT for variable image resolutions. Prior methods like D2NWG and SANE flatten weights into fixed-size representations and thus cannot handle this.**
>
> The combination of GHN alignment and patch tokenization is what enables width generalization, and neither component alone is sufficient. Destroying GHN's spatial structure via random neuron permutations collapses NNiT from 78.8\% to 0\% (see [W-3] below), while D2NWG trained on the same GHN data still degrades severely on unseen architectures without patch tokenization (Table 2).
>
> **[W-2]**
>
> >``The experiments primarily focus on relatively simple MLP strategies (ManiSkill3 task).''
>
> Following this suggestion, we evaluated NNiT on **MNIST digit classification** using the identical pipeline with no modifications. NNiT achieves **96.1\%** mean test accuracy across 80 policies from 8 unseen width combinations, with every policy above 80\%. This confirms the representation transfers across task families without pipeline changes.
>
> We evaluate on MLPs because they expose unconstrained per-layer width variation, a strictly harder setting than the fixed-per-stage-width CNN benchmarks where prior methods operate. Existing benchmarks vary depth with fixed per-stage widths (e.g., ResNet-20/32/44/56), a regime D2NWG performs reasonably well on. In an MLP, each hidden layer independently takes any width from $\{16,32,64\}$, producing $3^4 = 81$ topologies with unconstrained per-layer widths, making MLPs the natural minimal setting to study this problem. Extending GHN alignment and patch tokenization to other weight tensor shapes remains an important direction for future work.
>
> **[W-3, (Question) Q]**
>
> >``Lacks corresponding experimental results to verify zero-shot generalization... Can the method be applied to standard SGD-trained networks?''
>
> Table~2 in the paper provides direct evidence: NNiT receives only a held-out architecture token sequence (e.g., $[16,64,32,32]$), generates a full set of weight matrices, and assembles the corresponding MLP with zero fine-tuning, yielding 78.8\%/74.7\%/54.9\% mean success on 8 unseen architectures across different tasks. The mechanism is compositional: each width token from $\{input, 16,32,64, output\}$ maps to a learned patch count, and novel token sequences produce valid but previously unseen patch-count arrangements.
>
> We further isolate *why* this works through a controlled ablation. We apply random neuron permutations to GHN-generated weights, which preserve all weight statistics but destroy the 2D spatial ordering. Training NNiT on these permuted weights collapses performance from 78.8\% to 0\% (0/80 policies on PickCube-v1). Critically, this ablation closely approximates SGD-trained weights: cross-seed magnitude correlation drops to $r \approx 0.00$ for both permuted GHN and SGD-trained weights (vs. $r \approx 0.65$ for intact GHN weights). So, we modeled SGD-style weight populations while maintaining the same weight distributions and showed that NNiT cannot learn from SGD-style weight populations; the spatial structure provided by the GHN is essential for it to work.

---

> > ### Author Rebuttal · Reviewer_VtiK · 2026-04-05
> >
> > My concerns are solved to some extent.

---

> > > ### Author Response · Authors · 2026-04-06
> > >
> > > Thank you for your comments. We're glad to see that our clarification and additional study have addressed your questions. We will revise our paper accordingly and appreciate your advice in helping us improve the manuscript.
> > > Thank you again for your review.
> > > Authors

---

### Decision · Program_Chairs · 2026-04-30

**Decision:**

Accept (regular)

**Comment:**

All four reviewers converged on weak accept, and the rebuttal adequately addressed most concerns. The problem of width-agnostic weight generation is genuinely novel and worth exploring. Using GHNs seems like the correct tool. However, I share reviewers' concerns about the narrow empirical scope, and I find the justification for using CNN decoders in the GHN insufficiently explored. Why CNNs specifically? How do other decoder architectures (e.g., MLP, or locally connected networks, or RNNs, or attention-based decoders) affect the spatial alignment property that the entire method depends on? This is the key to the approach and deserves a more thorough investigation and explanation than what is currently provided. I ask the authors to address this with an ablation study in the camera-ready, along with incorporating the rebuttal experiments and tightening the framing to match the actual demonstrated scope.